# Expectancy-related changes in firing of dopamine neurons depend on hippocampus

Zhewei Zhang [1,3] ✉, Yuji K. Takahashi[1,3], Marlian Montesinos-Cartegena[1], Thorsten Kahnt [1], Angela J. Langdon [2,4] & Geoffrey Schoenbaum [1,4] ✉

The orbitofrontal cortex (OFC) and hippocampus (HC) both contribute to the cognitive maps that support flexible behavior. Previously, we used the dopamine neurons to measure the functional role of OFC. We recorded midbrain dopamine neurons as rats performed an odor-based choice task, in which expected rewards were manipulated across blocks. We found that ipsilateral OFC lesions degraded dopaminergic prediction errors, consistent with reduced resolution of the task states. Here we have repeated this experiment in male rats with ipsilateral HC lesions. The results show HC also shapes the task states, however unlike OFC, which provides information local to the trial, the HC is necessary for estimating upper-level hidden states that distinguish blocks. The results contrast the roles of the OFC and HC in cognitive mapping and suggest that the dopamine neurons access rich information from distributed regions regarding the environment's structure, potentially enabling this teaching signal to support complex behaviors.

The orbitofrontal cortex (OFC) and hippocampus (HC) are both implicated in learning the states, and relationships between them, that define the world around us[1–3]. These so-called cognitive or task maps are fundamental to the process by which we predict impending events, particularly valuable outcomes[4,5]. Both areas are also well-positioned to provide information, either directly or indirectly, to the midbrain dopamine neurons, which generate teaching signals that reflect discrepancies between actual and expected outcomes[6–8], i.e., prediction errors.

Several years ago, we capitalized on this arrangement by using the dopamine neurons as a sensor or tool to measure the functional effects of OFC lesions[9]. We recorded midbrain dopamine neurons as rats performed an odor-based choice task, in which errors in the prediction of reward were induced by manipulating the number or timing of the expected rewards across blocks of trials. We found that OFC lesions ipsilateral to the recording electrodes caused prediction errors represented by dopamine neurons to be degraded. In particular, the response to reward omission was largely abolished, while the response to unexpected reward was both diminished and slower to adapt with learning across trials. Activity to the predictive cues was also less closely tied to high and low values. Critically, these results were not

consistent with our initial hypothesis that the OFC provided information about the value of expected outcomes, and instead a computational modeling approach showed it was best understood as a loss in the resolution of the task states, particularly under conditions where hidden information was critical to sharpening the predictions. Notably, these data led to the hypothesis that the OFC is critical for representing the current state in the cognitive map, which is relied on by other brain regions, like midbrain dopamine neurons[2].

HC is another critical brain region implicated in encoding the task state, however the distinct contributions of the HC and the OFC to the cognitive map remain unclear. Early research indicates that HC neurons encode animals' current location[1,10] and guide goal-directed navigation[11–13]. HC is also crucial for episodic memory[14,15], and HC neuron activity differs across various contexts[16,17], reflecting the situation or environment in which a series of events occur. More recent studies suggest that the HC is involved in organizing memories within context. The replay or reactivation of HC neurons integrates information essential for understanding the current context[18–21], which can be triggered by behavioral events, such as the completion of a trial or receiving a reward[22,23].

[1]Intramural Research Program, National Institute on Drug Abuse, Baltimore, MD, USA. [2]Intramural Research Program, National Institute on Mental Health, Bethesda, MD, USA. [3]These authors contributed equally: Zhewei Zhang, Yuji K. Takahashi. [4]These authors jointly supervised this work: Angela J. Langdon, Geoffrey Schoenbaum. ✉e-mail: zhewei.zhang@nih.gov; geoffrey.schoenbaum@nih.gov

Here we have repeated the experiment described above[9], along with computational modeling of the results, to test the hypothesis that HC also provides critical information to the dopamine neurons regarding the layout of the task space. The results show that this is indeed the case, however unlike the OFC, which provides information local to the trial, the HC appears to be necessary only for estimating the upper-level hidden states, which are relevant to distinguishing the trial blocks. The results demonstrate the respective roles of the OFC and HC in cognitive mapping and how these areas interact to support learning. Additionally, they add to evidence that the dopamine neurons access a rich information set from distributed regions regarding the predictive structure of the environment, potentially enabling this powerful teaching signal to support complex learning and behavior.

## Results

We recorded single-unit activity in male Long-Evans rats with ipsilateral sham ($n = 5$) or neurotoxic ($n = 9$) lesions targeting the HC and resulting in visible loss of neurons in 54% (49-60%) of this region across subjects (Fig. 1). Activity was recorded in an odor-guided choice task identical to one previously used to characterize signaling of reward predictions and reward prediction errors in male Long-Evans rats with ipsilateral OFC lesions[9]. On each trial, rats sampled one of three different odor cues at a central port and then responded at one of two adjacent fluid wells (Fig. 1c). One odor signaled the availability of sucrose reward only in the left well (forced left), a second odor signaled sucrose reward only in the right well (forced right), and a third odor signaled the reward was available at either well (free choice). These same three odors were consistently used throughout the entire study, with their corresponding associations with actions unchanged. To induce errors in reward prediction, we manipulated either the timing or the number of rewards delivered in each well across 4 blocks of trials (Fig. 1d). The switches between blocks were not explicitly signaled. Positive prediction errors were induced by making a previously delayed reward immediate (Fig. 1d, $2^{sh}$ and $1^{st}$ bolus in $3^{bg}$) or by adding more reward (Fig. 1d, $2^{nd}$ bolus in $3^{bg}$ and $4^{bg}$), whereas negative prediction errors were induced by delaying a previously immediate reward (Fig. 1d, $2^{lo}$) or by decreasing the number of rewards (Fig. 1d, $4^{sm}$). Either well 1 or well 2 in Fig. 1d could be on the left side, and the other was on the right side, which was counterbalanced between sessions. Rats in both groups changed their choice behavior across blocks in response to the changing rewards, choosing the higher value reward more often on free-choice trials (Fig. 1e) and responding more quickly (Fig. 1f) and accurately (Fig. 1g) on forced-choice trials when the earlier or larger reward was at stake. As expected, there were no effects of the ipsilateral lesions on free choice trials on these behaviors (statistics in figure captions).

Units were recorded in the lateral VTA (Fig. 2a, insets, and we identified putative dopamine neurons by means of a waveform analysis like that used to identify dopamine neurons in primates[6,24–30]. This analysis isolates neurons in rat VTA whose firing is sensitive to intravenous infusion of apomorphine or quinpirole, agents known to selectively inhibit activity of midbrain dopamine neurons, and which are selectively eliminated by infusion of a Casp3 neurotoxin (AAV1-Flex-TaCasp3-TEVp) into VTA of TH-Cre transgenic rats[31].

This approach identified as putatively dopaminergic 72 of 390 (18.4%) and 117 of 510 (22.9%) neurons recorded from VTA in control and HCx rats, respectively (Fig. 2a). These proportions did not differ between groups (Chi-square = 2.67, $p = 0.102$) nor were there any effects of lesions on the waveform characteristics of these neurons (Fig. 2b, ANOVA (Group x waveform), $F_{1,187} = 1.45$, $p = 0.23$). Of these, 44 neurons (61%) in control and 66 neurons (56%) in HCx rats increased firing in response to reward (Fig. 2c, t-test, $p < 0.05$, compared with a 400 ms baseline taken during the inter-trial interval before trial onset). Average baseline activity did not differ significantly between the two groups for these neurons ($F_{1,108} = 1.05$, $p = 0.31$) as well as for the non-

responsive dopamine neurons ($F_{1,77} = 0.03$, $p = 0.82$) and the neurons that were classified as non-dopaminergic ($F_{1,709} = 1.45$, $p = 0.23$).

### Dopamine neurons signal prediction errors in response to manipulations of reward in controls

Prediction error signaling was readily observed in response to changes in reward in dopamine neurons recorded in control rats. The activity of these neurons was elevated in response to delivery of an unexpected reward and suppressed in response to an omission of expected reward (Fig. 3a). To quantify these changes, we compared firing in the first five and last five trials with two-tailed t-tests and found 13 and 7 out of 44 neurons showing significant changes for the positive and negative prediction errors, respectively. We also computed difference scores for each neuron by comparing the average firing at the beginning versus the end of the blocks at the time of reward delivery or omission. The distributions of these scores were shifted above zero when unexpected reward was delivered (left in Fig. 3c) and below zero when expected reward was omitted (right in Fig. 3c), reflecting that changes in firing (elevation or suppression) were maximal at the start of the block and then diminished with learning of the new contingencies as the block proceeded (Fig. 3d). These observations were consistent when responses induced by changes in timing or the number of rewards delivered were analyzed separately. (Supplementary Fig. 1a)

### Ipsilateral hippocampal lesions disrupt changes in reward-evoked activity in VTA dopamine neurons in response to reward manipulations

HC lesions had a marked effect on the changes in the firing of dopamine neurons caused by changes in reward. In particular, dopamine neurons recorded in rats with ipsilateral HC lesions did not increase firing when reward was delivered unexpectedly (left in Fig. 3b) nor did they suppress firing when an expected reward was omitted (middle in Fig. 3b). Only 5 and 2 out of 66 neurons showed significant changes for the positive and negative prediction errors, respectively, numbers which were significantly less than what was found in the sham rats (Chi-square test, $p = 0.0023$ and 0.016 for positive and negative prediction errors, respectively). These effects were also quantified by analyzing the difference scores between early versus late trials in relevant blocks. The difference scores were not different from zero when an unexpected reward was delivered (Fig. 3e left) or when an expected reward was omitted (Fig. 3e right), regardless of whether the prediction error was induced by changes in timing or the number of rewards (Supplementary Fig. 1b), reflecting the relatively flat firing in early and late trials of each type (Fig. 3f).

Consistent with these apparent differences, a direct comparison of the data from control and HCx rats (ANOVA, group x reward/omission x early/late x trial, Fig. 3d versus Fig. 3f) revealed significant group interactions (group x reward/omission x early/late; $F_{4,432} = 13.4$, $p = 2.65e\text{-}11$), and the distributions of the difference scores comparing firing changes in reward delivery or omission (histograms, Fig. 3c versus 3e) were significantly different between the group (Wilcoxon rank-sum test; reward delivery, $p = 0.043$; reward omission, $p = 0.0005$). Thus dopamine neurons recorded in rats with ipsilateral HC lesions show degraded bidirectional changes in firing—presumed to be reward prediction errors—in response to manipulations of reward.

### Ipsilateral hippocampal lesions do not disrupt changes in cue-evoked activity in VTA dopamine neurons in response to reward manipulations

The activity of dopamine neurons in control rats also differed during sampling of the odor cues according to the expected value of the cue in the different blocks. On forced-choice trials, these neurons exhibited higher firing during the presentation of the high-valued cue than during presentation of the low-valued cue, a difference that reversed in

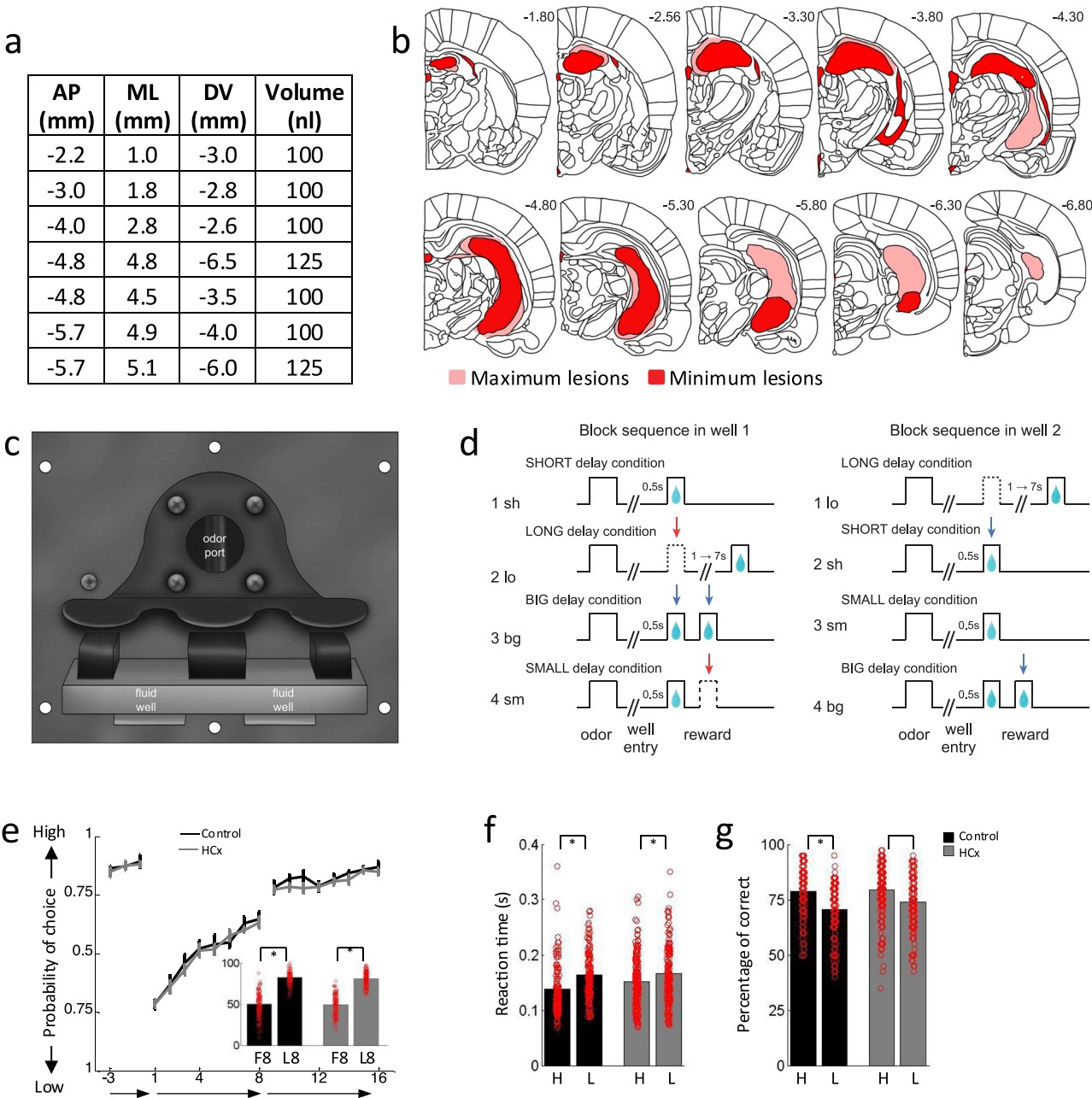

**Fig. 1 | Lesions, task design, and behavior. a** Table gives volumes and coordinates (AP and ML relative to bregma and DV relative to brain surface) of injections. **b** Brain sections illustrate the extent of the maximum (light red) and minimum (dark red) lesion at each level in HCx in the lesioned rats. **c** Picture of apparatus used in the task, showing the odor port (~2.5 cm diameter) and two-fluid wells. **d** Schematic of task design. Deflections indicate the time course of stimuli (odor and reward) presented on each trial. Dashed and solid lines show when a reward was omitted and delivered, respectively. Blue arrows, unexpected reward delivery; Red arrows, unexpected reward omission. **e** Choice behavior in free-choice trials before and immediately after the block switch and at the end of the subsequent block. Bar graphs indicate average percentage of choice for high-valued reward in first 8 and last 8 trials after block switch. In both groups, rats chose high valued well more often on later trials than earlier trials (Control, $p = 2.0e\text{-}4$; HCx, $p = 1.1e\text{-}8$). Three-way ANOVA comparing Group x Early/Late in choice revealed a significant main effect of value ($p = 1.0e\text{-}11$), but there were no

significant main effect of Group ($p = 0.41$) nor significant int[70]eraction of Group x Early/Late ($p = 0.70$). **f** Reaction time in response to high and low valued reward on last 10 forced trials across all blocks. Both groups showed faster reaction time when the high valued reward was at stake (Control, $p = 0.021$; HCx, $p = 0.013$). **g** Percentage of correct in response to high and low valued reward on last 10 forced trials across all blocks. Both groups showed higher accuracy when the high-valued reward was at stake (Control, $p = 0.019$; HCx, $p = 0.0025$). Three-way ANOVA comparing Group x Value revealed a significant main effect of value in reaction time ($p = 2.6e\text{-}4$) and in percentage of correct ($p = 6.5e\text{-}5$), but there were no main effect of Group (reaction time, $p = 0.77$; percentage of correct, $p = 0.73$) nor significant interaction of Group x Value (reaction time, $p = 0.16$; percentage of correct, $p = 0.19$). Data in (e-g) are presented as mean values +/− S.E. * in (**f**–**g**) represents the $p < 0.05$ from ANOVA. No adjustments were made for multiple comparisons.

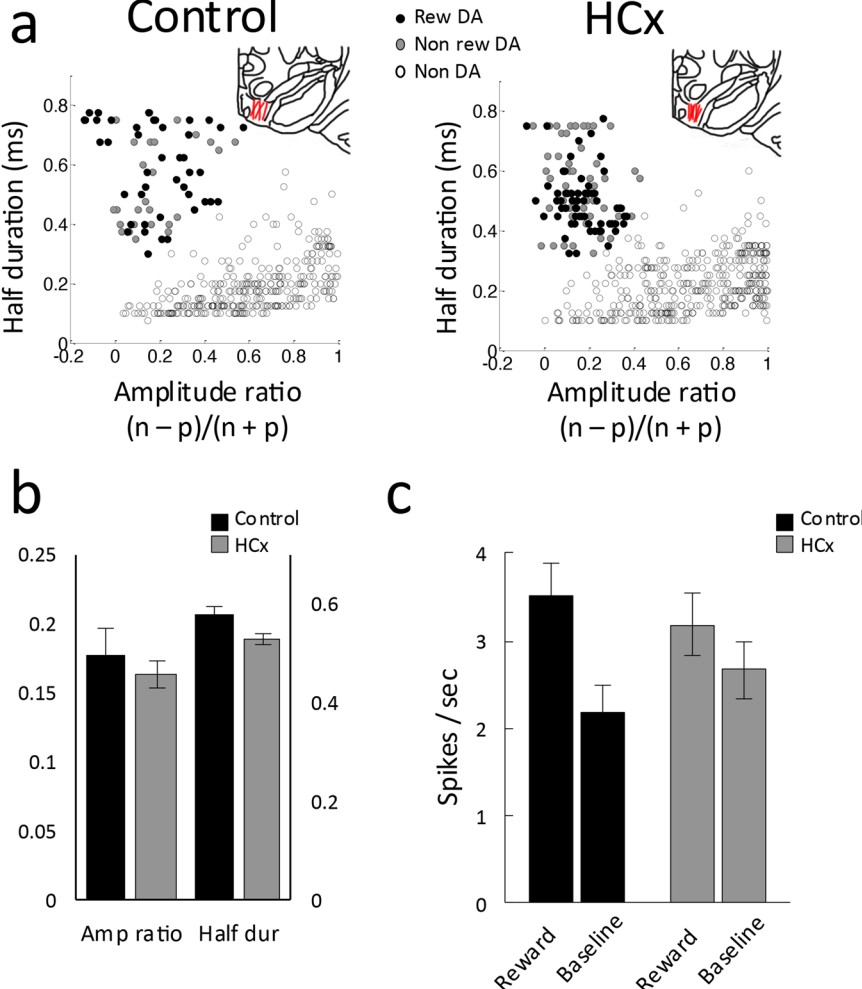

**Fig. 2 | Characterization of dopamine neurons in control and HCx rats. a** Results of cluster analysis based on the half time of the spike duration and the ratio comparing the amplitude of the first positive and negative waveform segments ($(n - p)/(n + p)$) in control (left) and HCx (right) groups. Reward responsive dopamine neurons (Rew DA), reward non-responsive dopamine neurons (Non rew DA), non dopamine neurons (non DA). Insets in each panel indicate location of electrode tracks in VTA in red for control (left) and HCx (right) rats. **b** Bar graphs indicating average amplitude ratio and half duration of putative dopamine neurons in control (black, $n = 72$) and HCx (gray, $n = 117$) groups. **c** Average firing of putative dopamine neurons during reward epochs versus similar 400 ms baseline period taken during the intertrial interval in control (black, $n = 72$) and HCx (gray, $n = 117$) groups. Error bars, S.E.M. 2-way ANOVA comparing group (control/HCx) x epoch (reward/baseline) revealed a significant main effect on epoch ($p = 4.4e-16$) and a significant interaction between group x epoch ($p = 7.4e-5$).

each block early in learning (Fig. 4a). On free-choice trials, the firing during the presentation of the free-choice cue reflected the more variable option (Supplementary Fig. 2a). To quantify how the cue-evoked activity changed over trials, we computed the difference scores comparing each neuron's firing to the high- versus low-value cues in early versus late trials. Distribution of these scores was shifted significantly above zero (Fig. 4b) in controls, indicating an increase/decrease in activity to the high/low value cues across trials, respectively.

Surprisingly, largely similar firing changes were evident in dopamine neurons recorded in the HC-lesioned rats, particularly on forced choice trials (Fig. 4c, d). A direct comparison of the data between control and lesioned groups (group x value x early/late x trial) revealed a significant main effect of value ($F_{1,108} = 19.8$, $p = 2.1e-5$) and significant interactions between value x early/late ($F_{1,108} = 12.7$, $p = 5.5e-4$), value x trial ($F_{4,432} = 3.87$, $p = 0.0042$) and value x early/late x trial ($F_{4,432} = 2.64$, $p = 0.33$). However, there were no significant main effects nor interactions involving group (F's < 1.5, p's > 0.10). Thus, dopamine neurons recorded from rats with ipsilateral HC lesions showed normal changes in firing in response to presentation of the differently-valued cues on forced choice trials. The only exception to this was in activity

to the free-choice cue in later trials, which was significantly lower than that to the high-valued cue (ANOVA, $F_{1,65} = 10.24$, $p = 0.002$) and significantly higher than that to the low-valued cue (ANOVA, $F_{1,65} = 6.85$, $p = 0.011$; Supplementary Fig. 2b).

## Hippocampal lesions disrupt hierarchical segregation of states available in different blocks

The neural results show that HC is necessary for normal error signaling by VTA dopamine neurons; dopamine neurons recorded in rats with ipsilateral HC lesions failed to signal prediction errors to changes in reward. However, the same neurons showed roughly normal error signals to the presentation of the predictive cues in our task. To better understand what hippocampus might contribute to this surprising pattern of results, we developed two different temporal-difference reinforcement learning models to describe the task and then monitored the error output of the models caused by changes in several parameters in an attempt to recreate these neural findings—one model inspired by our prior results on the effects of OFC lesions and a second model inspired by the failure of this model, which employed a state space better reflecting the complexity of the task, particularly the block structure.

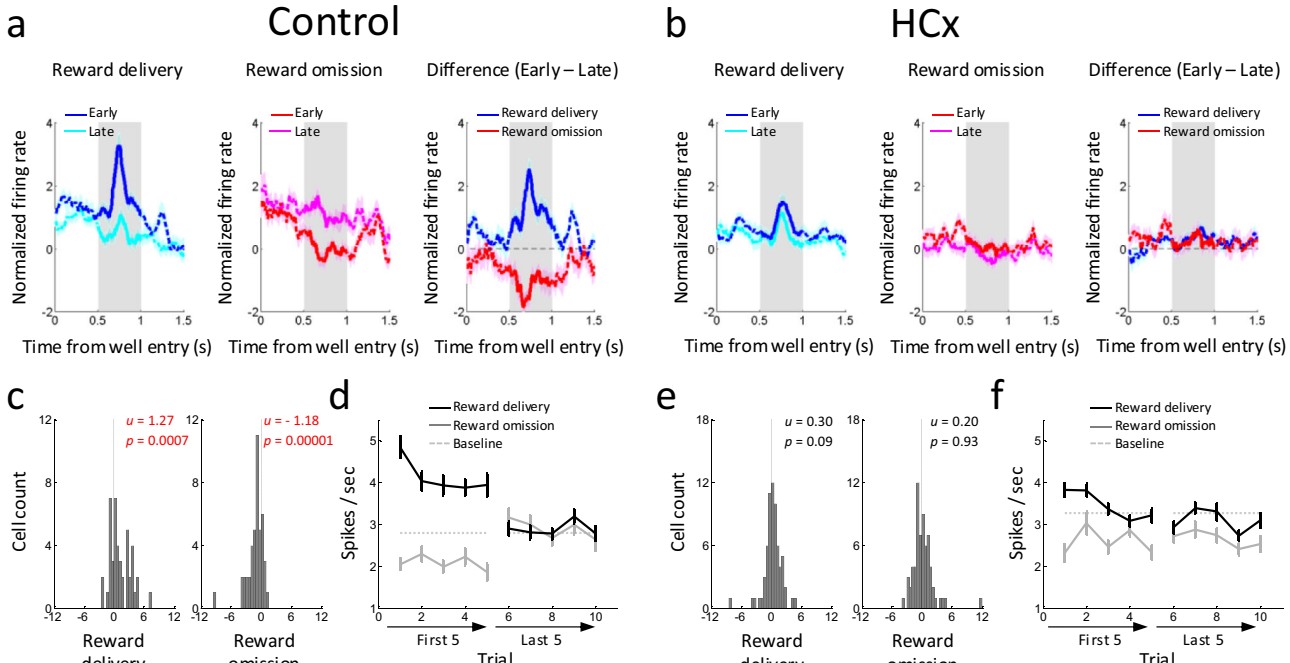

**Fig. 3 | Changes in activity of reward-responsive dopamine neurons to changes in reward value. a, b** Population responses of reward-responsive dopamine neurons in Control (**a**) and HCx (**b**) groups. Left panels show changes in firing to reward delivery on the first (dark-blue) and last (light-blue) trials. Middle panels show changes in firing to reward omission on the first (dark-red) and last (light-red) trials. Right panels show the difference in firing between first and last trials in response to reward delivery (blue) and omission (red). **c, e** Distributions of difference scores comparing firing to unexpected reward delivery (left) and omission (right) in the early and late trials in control (**c**) and HCx (**e**) groups. The numbers in each panel indicate results of two-sided Wilcoxon singed-rank test (*p*) and the average difference score (u). **d, f** Average firing in response to reward delivery (black) and omission (gray) in the first 5 and last 5 trials in control (**d**) and HCx (**f**) groups. ANOVA (Reward x Early/Late x Trial) revealed significant main effects of Reward (Control, *p* = 8.0e-4; HCx, *p* = 9.8e-6) and Trial (Control, *p* = 0.039; HCx, *p* = 5.8e-4) in both control and HCx, and a significant interaction of Reward x Early/Late in control ($F_{4,172}$ = 44.1, *p* = 1,1e-16), but not in HCx (*p* = 0.10). A step down in each plot revealed a significant main effect of Early/Late in reward delivery in both groups (control, *p* = 5.3e-6; HCx, *p* = 0.031) and reward omission in control (*p* = 3.9e-4), but not in HCx (*p* = 0.77). Significant effects are highlighted in red. Dashed lines indicate the baseline firing. Error bars, S.E.M.

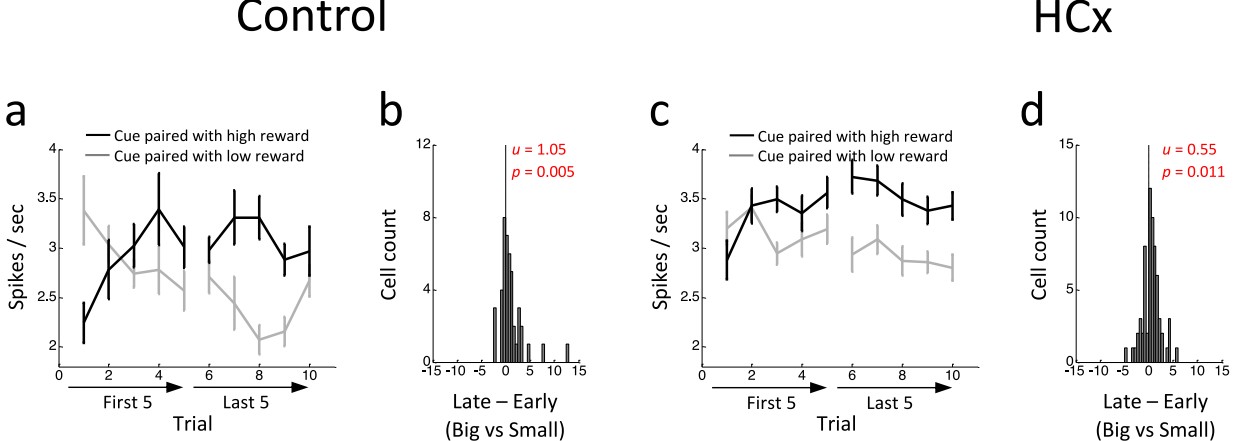

**Fig. 4 | Changes in reward-evoked activity of reward-responsive dopamine neurons to reward predictive odor cues. a, c** Average firing in response to high- (black) and low-valued (gray) cues in the first 5 and last 5 trials in control (**a**) and HCx (**c**) groups. ANOVA (group x value x early/late x trial) revealed a significant main effect of value (*p* = 2.1e-5) and significant interactions of value x early/late (*p* = 5.5e-4), value x trial (*p* = 0.0042), and value x early/late x trial (*p* = 0.033). Error bars, S.E.M. **b, d** Distributions of difference scores between high- and low-valued cues in early and late trials in control (**b**) and HCx (**d**) groups. The numbers in each panel indicate results of two-sided Wilcoxon singed-rank test (p) and the average difference score (u). Significant effects are highlighted in red.

In both models, the learning agent represented the behavioral task as serial transitions between hidden states within a partially-observable semi-Markov process[32]. States were probabilistically associated with observations, which mark the start of the time spent in that state, known as dwell time. While observations unambiguously signaled transitions between states, transitions could also happen without observations. The semi-Markov process differs from a Markov process in that the dwell time in each state has a unique probability distribution. As in other partially-observable settings, the belief state, a probability distribution over state occupancy, tracked the agent's

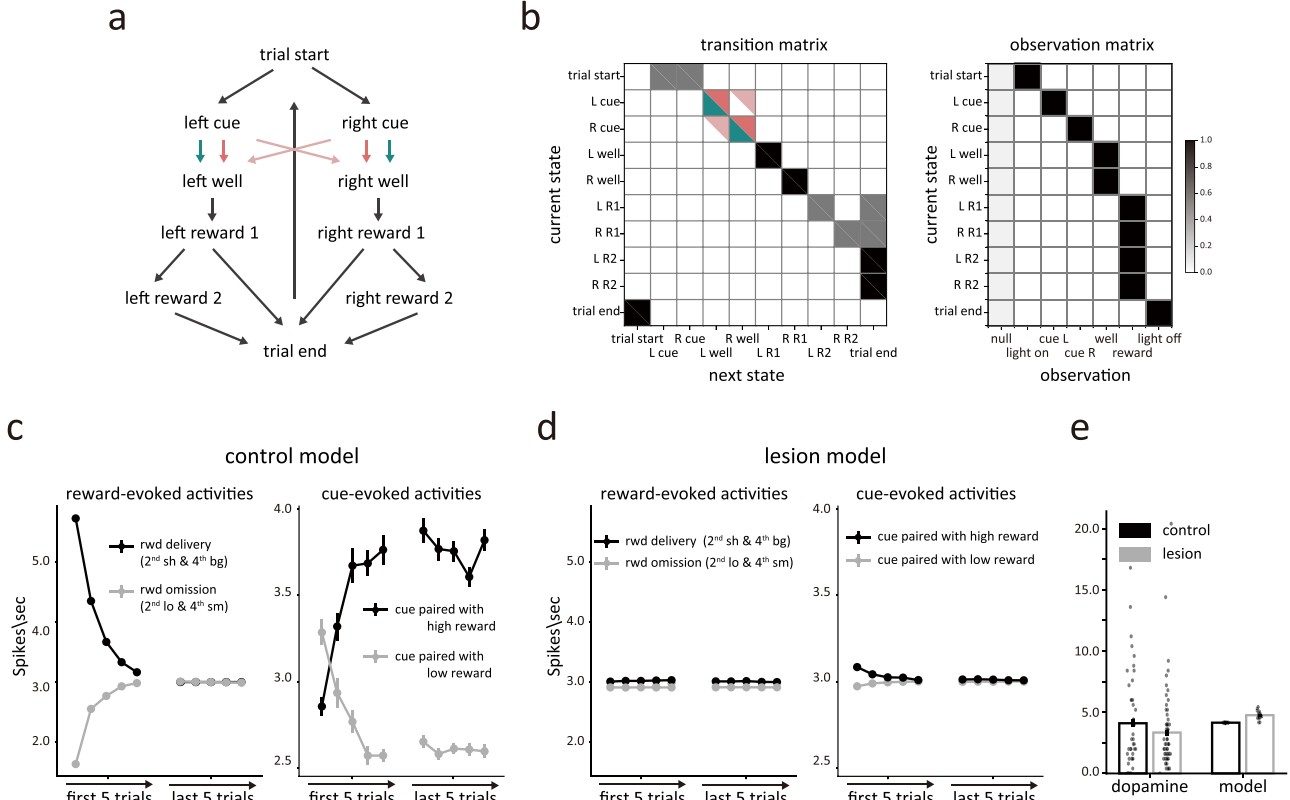

**Fig. 5 | Modeling the effect of hippocampal lesions as a blurring of transitions between trials. a** Conventional state space representation of the task. Possible transitions between states are depicted by arrows. Green arrows represent transitions available to the control model, while pink arrows represent transitions available to the lesion model. **b** The transition matrix (left panel) shows the probability of each successor state given each state, and the observation matrix (right panel) shows the probability of each observation given each state. The darker color indicates higher probabilities. Green and pink indicate the transitions available to the control and lesioned models, respectively. The characteristic observation is emitted with $p = 0.95$. States also emit a null (empty) observation ($p = 0.05$) or any of the other five possible observations (with $p = $ 1e-4 each). The observation probabilities for each state were normalized by dividing their sum. **c** Simulated average prediction errors in the control model during the 2nd and 4th blocks. In the left panel, the black and gray lines represent the prediction error in response to reward delivery and reward omission, respectively. In the right panel, the dark and light lines represent the prediction error in response to the odor cue paired with high and low reward, respectively. **d** The same format as Fig. 5c, but for the lesion model. **e** Comparison of activities evoked by the 2nd drop of reward in the first 5 trials of the 3rd block in both animals and the model with a flat task space. The dopamine neurons in sham animals ($n = 44$) exhibit a higher response compared to the HCx animals ($n = 66$). In contrast, the model with a flat state space shows the opposite pattern ($n = 20$). Data in (**c**–**e**) are presented as mean values +/− S.E.

understanding of their current position within the various possible states of the task[33]. To estimate the current belief state, the agent combines several factors: task observations (i.e., number of rewards), the expected dwell time in the current state (i.e., whether short or long), and their knowledge of the task's structure (i.e., transitions between states and the likelihood of observations and dwell time). When the agent inferred that a state transition had occurred, it calculated a reward prediction error and updated the value of the previous state through a temporal-difference learning rule, and, in parallel, the dwell-time distribution was updated to reflect the time spent in that state.

Using this basic architecture, we developed two different models to explain the effects of HC lesions. The first model was simpler, reflecting only the states and dwell-time distributions the rats experienced when performing the task (FigS. 5a, b, model 1). This model's state space was designed to be flat, simply reflecting rats' physical locations and observations during the task, without any additional levels of organization related to how the trials were blocked or other factors. The state space consisted of ten states: trial start, left/right cue, left/right well, left/right rewards (1st and 2nd drop), and inter-trial interval, as illustrated in Fig. 5a. The transitions between states were governed by the transition matrix shown in Fig. 5b (left panel). Observations, i.e., light onset/off, odor cues for left/right choices, rewards, and a null observation, indicated the transition to different

states. The observation matrix showed the probability of each observation given each state (Fig. 5b, right panel). In this model, the agent learned the value and dwell times of each state through experience to minimize the deviation between expectations and observations. This resulted in a pattern of reward prediction errors at the time of reward delivery and cue sampling very similar to that observed in dopamine neurons recorded from control rats (Fig. 5c).

To model the effects of HC lesions, we blurred the ability of the model to maintain internal information about the transition probabilities between the odor cues and the correct well states. This was achieved by increasing the probabilities of transition from the left cue state to the right well states, and from the right cue state to the left well state, from baseline ($10^{-4}$) to 0.45 (pink color in Fig. 5a, b). This reflected the hypothesis that HC lesions would prevent the brain regions responsible for state estimation from maintaining the hidden information during the period after a response had been made, while the rats were waiting for reward, an effect, essentially like that caused by lesions of OFC in this task[9]. In the context of this model, this change caused the agent to rely more heavily on external observations and dwell time than on the transition matrix when estimating the current state. As a result, the agent was more likely to respond to unexpected external input regarding events or timing− like the unexpected appearance of reward (Fig. 1d, $2^{sh}$ or $4^{bg}$) or its unexpected delay or omission (Fig. 1d, $2^{lo}$ or $4^{sm}$)−by changing its estimate of the current

state to the opposite well (Fig. 5b, light pink shading). This re-evaluation resulted in the loss of prediction error signaling by the agent in these blocks, since it essentially brought their belief state and its predictions into alignment with external events (Fig. 5d, left panel; a detailed example was presented in Supplementary Fig. 3). This result aligns well with main effects of lesions on dopamine neuron firing in the task – specifically the apparent loss of responsivity to unexpected reward and reward omission (Fig. 3).

However, this model predicted something that was not observed at the transition between blocks 2 and 3. Because block 3 is the first block in the session that involves two drops of reward, there is no internal state at the end of block 2 whose value estimate can predict the delivery of the second reward (Fig. 1d, 3$^{bg}$). As a result, the lesioned model produced a strong positive prediction error to the second reward on these trials (Fig. 5e). This effect was not evident in the activity of dopamine neurons recorded in the HCx rats (Fig. 5e). Further, the lesioned model also failed to produce the surprisingly normal error responses to the odor cues observed in the dopamine neurons recorded in HCx rats (Fig. 5d, right panel). On the contrary, the

lesioned model predicted nearly unchanged error responses to the odor cues. This discrepancy arose from the loss of error signals at the time of reward and no errors could be backpropagated to cues to update their values in the model.

Given the poor match between the output of the simplified lesion model and the data, we developed a second model, based on the same semi-Markov process but a more complex and more realistic state space that recognized the extensive training history of the rats on the task by creating separate clusters of states for each of the different blocks. Each cluster of states mimicked the states in the first model, but adapted to the unique reward contingencies of each block. The observation matrix from the first model was retained to control the probability of observation given states in each cluster (Fig. 5b). This resulted in a multi-level or hierarchical state space in which lower-level states (light blue box in the Fig. 6a) described the underlying process within individual trials and upper-level states (dark blue box in the Fig. 6a) provided priors of transitions to states that belonged to each block, which were updated based on recent reward history. The hier-archical state space predicted that rats should choose the high-valued

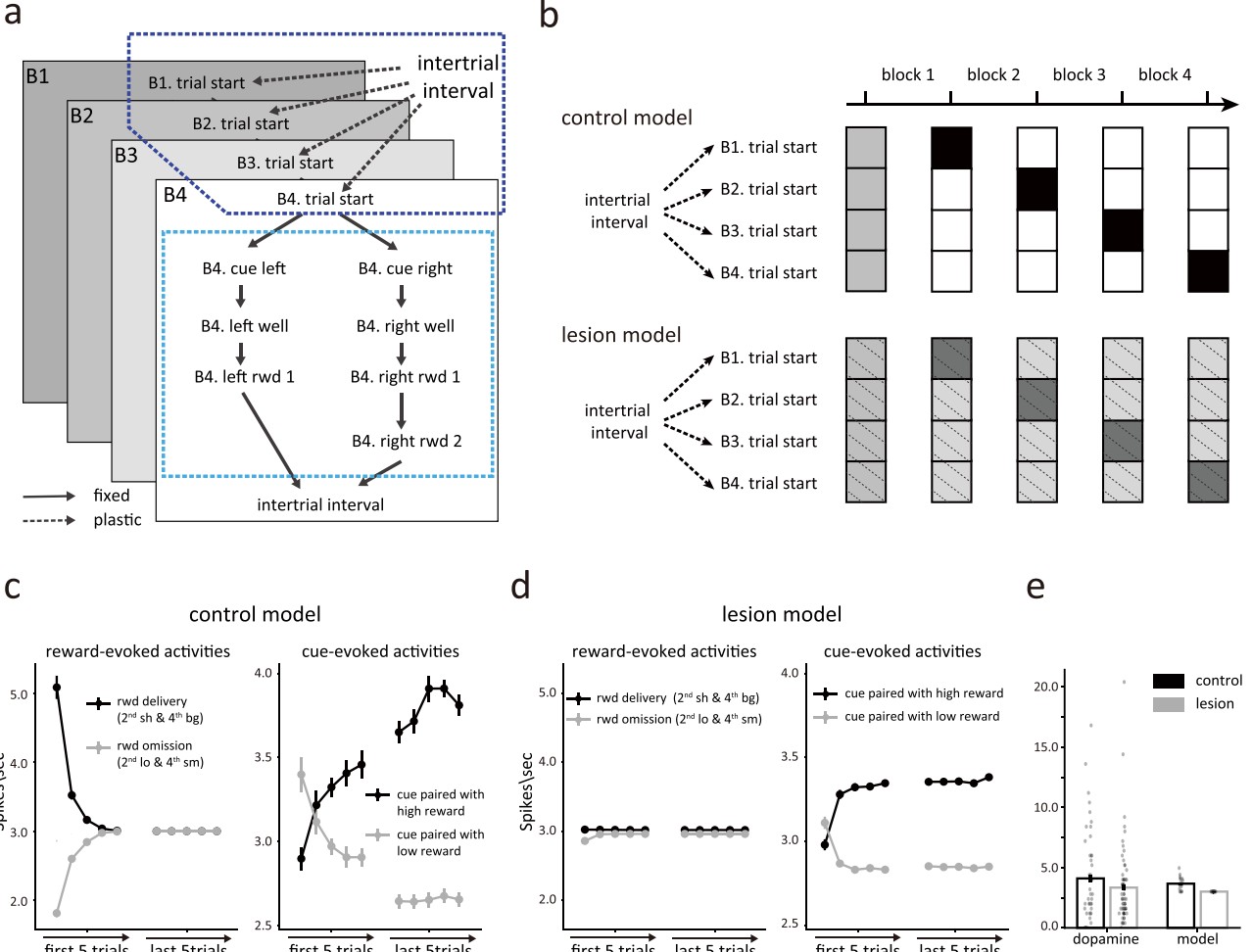

**Fig. 6 | Modeling the effect of hippocampal lesions as a blurring of transitions between blocks. a** Multi-level or hierarchical state space representation of the task. The upper-level (indicated by the dark blue box) contains the transition from intertrial interval to each block, with the trial start state in each block leading to the lower-level states (indicated by the light blue box), describing the state space of individual trials. Available transitions between states are marked by arrows. Dashed arrows indicated plastic transitions, whose probabilities are updated during learning. Solid arrows are transitions with fixed probabilities. **b** The transition probabilities among the upper-level states are updated according to reward history. The control model learns the transition probabilities perfectly and with low

uncertainty (top panel), while the lesioned model has greater residual uncertainty (bottom panel). The darker color indicates higher probabilities. **c** Simulated aver-age prediction errors in the control model during the 2$^{nd}$ and 4$^{th}$ blocks. In the left panel, the dark and light lines represent the prediction error in response to reward delivery and reward omission, respectively. In the right panel, the black and gray lines represent the prediction error in response to odor cues paired with high and low reward, respectively. **d** The same format as Fig. 6c, but for the lesion model. **e** Same format as Fig. 5e, but for the model with a hierarchical task space. The hierarchical model ($n = 20$) reproduces consistent error signal patterns with dopamine neuron activities. Data in **c**–**e** are presented as mean values +/− S.E.

option more frequently in subsequent free-choice trials after receiving a long/big reward, which provided sufficient information for estimating the current block, compared to after receiving a short/small reward. Our analysis confirmed this prediction (Supplementary Fig. 5). As with the simpler model, this model did a good job reproducing the pattern of reward prediction errors evident at the time of reward delivery and cue sampling in dopamine neurons recorded from control rats (Fig. 6c).

Using this more complex state space, we again modeled the effect of hippocampal lesions as a blurring of transitions, but this time between the upper-level states describing the trial blocks (Fig. 6b). This reflected the hypothesis that hippocampus might differ from OFC in that it would be necessary for maintaining hidden information relevant to identifying the trial blocks rather than regarding the recent response for state estimation. To implement this, we introduced larger uncertainty in the transition probabilities between the upper-level states by increasing the minimum value of the transition probabilities to each state to 0.15 (Fig. 6b). Though counterintuitive at first glance, the imprecise transition probabilities caused the lesioned model to again be more heavily influenced by external observations about events and dwell times, in this case affecting its estimate about the current block (a detailed example was presented in Supplementary Fig. 4). As a result, the lesioned model was again able to quickly alter its estimated state to states belonging to other blocks to adapt to changes in reward number or timing, rather than adjusting the state value through error signaling, but unlike the prior model, this ability now included the third block where two rewards were introduced (Fig. 6e).

This model also captured the relative preservation of error signals in response to the odor cues observed in the HCx rats (Fig. 4). This was possible because the lesioned model updated the transition probabilities based on reward history, leading to an update in the estimated belief state behind the odor cues after the initial trial in a block and resulting in changes in cue-evoked activity (Fig. 6d, right panel). Interestingly, the lesioned model also exhibited clear differences in the cue-evoked errors that arguably mirrored minor but apparent differences in the preserved responses in HCx rats; in particular, while not significant, the cue-evoked response in these rats appeared weaker and changed mostly after the initial trial whereas the change in controls is more gradual (Fig. 4). The second model showed similar features in cue-evoked firing when lesioned. The dopamine responses in free-choice trials were also captured by this model with hierarchical task space (Supplementary Fig. 1c, 1d; detailed methods and explanation can be found in the supplementary information).

To rule out the possibility that the distinct pattern of prediction errors between two models was from arbitrarily chosen parameters, we fitted models by maximizing the likelihood of observing dopamine neurons firing in vivo. Due to the simulation's complexity, we fitted three key free parameters in each model that we believe most significantly affect the prediction error. For the flat model, we adjusted the learning rate for value, the dwell distribution, and the transition probabilities from cue states to the well state on the same side in the lesion model. For the hierarchical model, we adjusted the learning rate for value, the transition probability, and the transition uncertainty in the lesion model. With the best-fitting parameters, we calculated the likelihoods for two models and compared them using a likelihood ratio test. A significantly higher likelihood ($p = 9.6e-31$, Chi-square test) was found in the second model with hierarchical task space, demonstrating it provides a more accurate explanation of the activity of dopamine neurons. Overall, this second model in which a HC lesion was based on blurring of upper-level internal information about the states available in different blocks effectively captured the characteristics of dopamine neurons' firing in lesioned rats, suggesting that HC may be particularly important for maintaining and updating higher-order state representations that capture the task block structure within the cognitive map.

## The lesions in the hippocampus, orbitofrontal cortex, and ventral striatum impact state representation in complementary ways

We next applied the same methodology to reexamine the effects of OFC and VS lesions on dopaminergic error signaling reported in previous studies[9,34]. To gain a deeper understanding of how the ipsilateral lesion of these brain areas affects dopaminergic prediction error signals and to ensure fair comparisons, we lesioned the hierarchical model in a manner similar to our prior work, where a flat task space was used, to reproduce the effects of OFC and VS lesions.

Dopamine neurons in rats with ipsilateral OFC lesions failed to suppress firing to reward omission and exhibited weaker but also more persistent increases in firing to unexpected rewards[9,34]. A thorough model comparison showed that the ambiguity between left and right well states hindered learning and resulted in weaker but more constant prediction errors, providing the best explanation for the experimental findings. To reproduce this effect, we modeled the effects of OFC lesions in the hierarchical model by blurring the model's ability to maintain internal information within each block, specifically affecting the transition after actions (Fig. 7a), which caused the model to be unable to differentiate the two wells or track the reward associated with each well. The resultant lesioned model reproduced the altered prediction errors in response to rewards observed in the prior study in rats with OFC lesions (Fig. 7c).

Dopamine neurons in rats with ipsilateral VS lesions exhibited intact prediction errors in response to changes in the number of rewards but showed no error signals in response to changes in reward timing on the order of several seconds[9,34], suggesting the normal temporal expectation was deficient due to the VS lesion. To reproduce this effect, we modeled the effects of VS lesions by preventing the model from accurately learning the dwell time spent in each state (Fig. 7b), which caused it to be unable to form precise temporal expectations or deduce unobservable state transitions. Consequently, there were no prediction errors when the reward timing changed or the reward was omitted but normal errors when additional rewards were delivered (Fig. 7d), consistent with the response in rats with VS lesions.

These results reveal complementary roles for the HC, OFC, and VS in constructing cognitive maps—at least in our task, in well-trained rats, and as referenced by prediction error signaling dopamine neurons. The OFC is necessary for differentiating between states following actions that lack clear external differences, like the left and right wells in this case. In contrast, HC is particularly essential for accurately representing the current context or trial block, which is also hidden or partially observable. Meanwhile, VS is required to track timing and to estimate the occurrence of unobservable transitions.

## Discussion

In the current study, we utilized the well-characterized dopaminergic reward prediction error as a tool to investigate the critical role of the HC in the development of cognitive or task maps for guiding learning from direct experience of rewards, repeating an approach used previously to characterize the role of OFC in cognitive mapping[2,9]. We recorded the activity of midbrain dopamine neurons during the induction of prediction errors in an odor-based choice task and compared activity in controls with the activity in rats with ipsilateral HC lesions. Combined with computational modeling, this allowed us to gain insight into the specific effects of HC removal on the structure of the internal task representation being used for learning and compare them to the effects of OFC lesions. The results show that HC, like OFC, contributes to the representation of the appropriate state structure in this task, at least with regard to the information available to midbrain dopamine neurons, and appears to be especially important for properly representing information that is hidden or at least partially observable. However, unlike OFC, this contribution is not required for

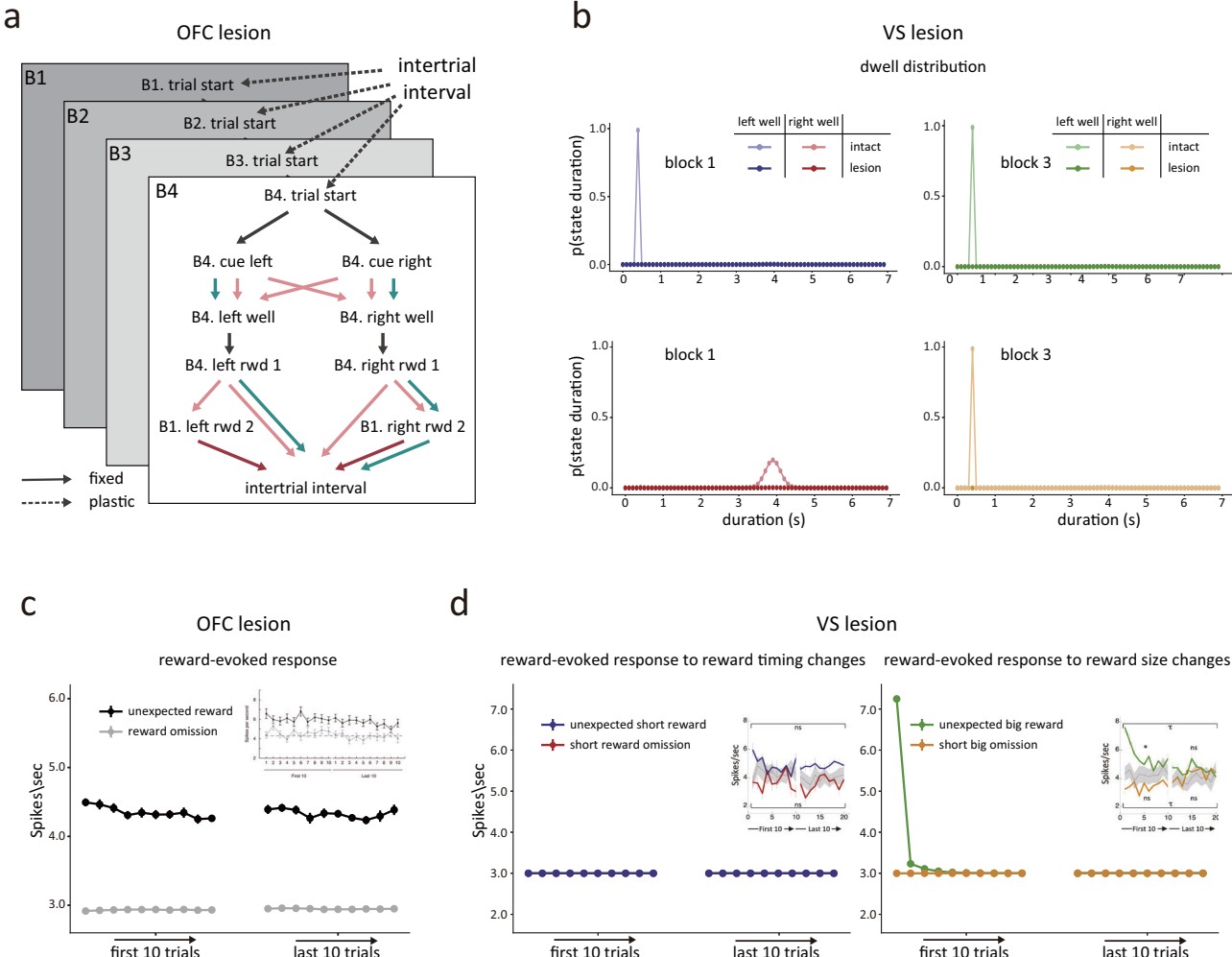

**Fig. 7 | Reproduce the changes in dopamine neuron firing after OFC or ventral striatal lesions. a**, **c** As indicated by (**a**), to simulate an OFC lesion in the model, we fully eliminated the ability of the model to differentiate between states after actions by blurring transition probabilities between the odor cues and the corresponding well states and allowing the reward 1 state transit to either reward 2 state and intertrial interval state for both well in the third and fourth block. Possible transitions between states are depicted by arrows, and the darker color indicates higher probabilities; green arrows represent transitions available to the control model, while pink arrows represent transitions available to the lesion model. The model can account for the dopamine neurons' response at the time of unexpected reward delivery (black lines, (**c**)) or omission (gray lines, (**c**)) in OFC lesion rats. Insert: Average firing of dopamine neurons after reward delivery (black lines) or omission

(gray lines) in rats with OFC lesions. (**b**, **d**) Left panel in (**b**): Dwell-time distributions learned at the end of block 1 for the state left well in the short delay condition (blue) and state right well in the long delay condition (red) for the intact and VS lesion models. Right panel in (**b**): Dwell-time distributions learned at the end of block 3 for the state left well in the big reward condition (green) and state right well in the small reward condition (orange). By preventing the model from learning and using precise dwell time staying in each state, the model fails to learn the dwell-time distribution (**b**) and deduce unobservable transitions, resulting in no prediction error when the short reward is delivered (blue lines, (**d**)) or omitted unexpectedly (red lines, (**d**) and yellow lines, (**d**)), but the prediction error to the unexpected big reward delivery (green lines, (**d**)) remains intact. These results are consistent with the response patterns observed in dopamine neurons after ventral striatal lesions (inserts).

properly segregating states within the trial but instead becomes important at longer timescales, across trials, where it is critical to organizing information about the context or current block.

In evaluating these results, it is worth noting that the models used in the current study are largely the same as the model used previously to understand the effects of OFC lesions in this task[9], aside from the addition of a semi-Markov process to allow the model to learn about reward timing and selectively disrupted by ventral striatal lesions[34]. The current models inherited the basic task architecture from the prior work on OFC, although the second model was extended with a HC-dependent hierarchical element to accommodate separate representations of the states available in different blocks of trials. Critically, by modifying either the transition matrix or dwell-time distribution in the basic model used here, we were able to replicate the changes in dopamine neuron firing observed in prior work after OFC or ventral striatal lesions. Thus, the divergent results here are not due to

idiosyncratic differences between the models and do not alter conclusions drawn in those prior studies.

While prior studies[35,36] have emphasized similarities in task space encoding between the HC and OFC, the current findings reveal important distinctions. These findings are significant both for contrasting the roles of HC and OFC in cognitive mapping as well as for revealing the complexity of information afferent to the midbrain dopamine neurons[37,38]. With regard to HC and OFC function, the results are consistent with the idea that both areas are critical for constructing maps of complex task spaces, but within the constraints of our task they point to different contributions. The OFC was particularly important for segregating states—in this case the left and right wells—significant to reward (i.e., one direction is rewarded, and one is not on forced-choice trials, and one is always better on free choice) and negligible external differences (both wells are identical). This function is consistent with evidence in other settings that the OFC is particularly

critical for representing latent, hidden, or partially observable information of potential behavioral relevance[2,39–41]. By contrast, the HC was not necessary for maintaining this information, however it played a key role in properly representing and updating the estimation of the current context or trial block. Information about the current trial block is also hidden or partially observable (i.e., there are no unique events that identify different blocks in the ITI or even during the trial until the actual reward), however properly maintaining such information has a strong temporal component requiring memory, thus the dependence on HC is clearly consistent with the historical role of HC in supporting episodic memory and contextual information[42–44], as well as more recent evidence that the HC encodes time as well as parsing statistical regularities that are critical to constructing hierarchical task spaces[45,46]. Specifically, updating the estimation of the trial block requires integrating information about odor identity and reward outcome across preceding trials. This process could be facilitated by the reactivation of HC neurons in patterns similar to those experienced during previous trials, which has been widely described in many previous studies[19,47–49]. This reactivation could replay the information over a long time window and be triggered by a salient stimulus or event boundary[37,38] that marks the end of one series of events and the beginning of another, such as rewards and intertrial interval[22,23]. Without an intact HC and the reactivation of HC neurons, the estimation of the block may become more uncertain. It is worth noting that while similar reactivation patterns have been observed in other brain regions, including the medial prefrontal cortex[50] and OFC[51], they seems to be insufficient to fully compensate for HC function.

Which one of these features is critical for the involvement of the HC in our task is difficult to say, however it is tempting to suggest it is the hierarchical, reward-orthogonal organization[42]. In complex environments with an overwhelming number of states, agents—humans or rats—are theorized to organize the cognitive map and plan actions hierarchically for efficient behavior[52]. Yet most behavioral tasks used in behavioral neuroscience are not obviously hierarchical, nor are the results analyzed to consider possible hierarchical solutions. This makes it difficult to ascertain whether the HC encodes states at all orders equally or exhibits greater encoding of upper-level states. Although our findings indicate that the HC lesion disrupts the clear delineation of transitions between upper-level states, we do not intend to suggest that the HC is not involved in encoding other lower-level states or their transitions. Rather, other brain regions, such as the OFC, may perform similar functions, allowing compensation.

Unilateral HC lesions did not result in significant behavioral changes, likely due to overtraining and the preservation of one intact hemisphere capable of maintaining normal behavior. This approach allowed us to investigate changes in DA firing resulting from HC output removal without behavioral confounds. Although we did not directly test the effects of bilateral hippocampal lesions on behavior in this study, previous research[53,54] has demonstrated that the hippocampus is crucial for context learning. Without it, animals can still learn but less efficiently. Consistent with these findings, our previous study[55], which manipulated reward size and type, found that rats with bilateral lesions of the hippocampal output area, the ventral subiculum, showed a reduced rate of choosing the higher-value reward in free-choice trials following block switches. without affecting choices in forced-choice trials.

Although we have shown the distinct contributions of the OFC and HC to the cognitive map, how they interact and contribute collectively remains a mystery. Our previous study[55] suggested that the inactivation of HC impaired the representation of task space in OFC. The current findings shed further light on this complex interaction. It appears that the estimation of the current block or upper-level state, potentially performed by HC neurons during the intertrial interval, depends on the lower-level states within a single trial, which might be estimated online by the OFC neurons. In turn, the estimation of the current upper-level state aids in the estimation of lower-level states. Thus, HC might utilize the lower-level states estimated by OFC to estimate the upper-level states and send this information back to OFC for further lower-level state estimation.

With regard to dopamine function, these results highlight the potential complexity of sources and information available to this powerful teaching system. VTA dopamine neurons receive information from numerous brain areas, both directly and indirectly[56]. Although these various inputs may be redundant when behavioral demands are low, the current results add to a growing body of data indicating that redundant coding may be limited to these simple situations, in which hidden structure is unimportant in determining the internal cognitive map according to which predicted values are calculated. In more complex situations, the unique contributions of different brain regions would be expected to—and clearly do— start to emerge. This points to the importance of studying the specific contributions of brain regions in behavioral contexts sufficiently complex to test their unique contributions.

Having access to rich sources of information regarding the task structure is also important because, though it is much ignored, the complexity of the internal state representation can have a dramatic influence on what sorts of behaviors can be supported by both the actual dopamine system and by the temporal difference learning rule which has been mapped onto it[57]. While much work in the field focuses on showing that current accounts either can or cannot explain certain findings, it is perhaps worthwhile considering whether much of the uncertainty or contradictory results could be explained by a better understanding of how individual animal subjects represent the state space of a task, along with the biases and idiosyncrasies that are native to volitional behavior. For instance, it has been suggested that contingency degradation provides an insurmountable challenge to the temporal difference reinforcement learning algorithms, since an agent armed with a very simple state space that considers only the cue will learn the value of a cue based on its contiguity with reward, without impact of non-contingent reward delivery during periods when the cue is not present[58,59]. However, if this agent has access to a state space in which the context is considered and allowed to be a target and compete for learning—including when the cue is present—then a temporal difference rule will in principle reproduce the known effects of contingency degradation (i.e., delivery of rewards during non-cue periods) on behavior and dopamine release to a degraded cue.

Indeed, the ability of the dopamine system to support learning and behavior in the real world, which is fantastically complicated compared to even our task, likely depends entirely on the complexity and detail of the input the system receives. A teaching signal is only as good as the information it receives about the world. This is increasingly being recognized by the incorporation of channels, features, and bases into temporal difference learning models[60–62] and in research positing that dopamine neurons are influenced by internal information, inference, and dynamically evolving beliefs[33,63–69]. Yet these findings are likely only the tip of the iceberg; the results here and in the related studies show experimentally that information from even high-level association cortices is utilized by these neurons.

## Methods

To allow direct comparisons to be made to prior work, all equipment and procedures used were substantially the same as those used in a prior study in which dopamine neurons were recorded in rats with ipsilateral lesions of the OFC[9]. In particular, the approach to recording dopamine neurons, the electrode design, recording systems, general task, specific task and training procedures, strain, age and sex of rat used were all identical to the prior study. All experimental methods were approved by the National Institute on Drug Abuse Intramural Research Program.

## Experimental Model and Subject Details

Data from fourteen male Long-Evans rats (Charles River Labs, Wilmington, MA) contributed to this study; this does not include two rats that expired post-operatively during recording whose data were not used. Rats were tested at the NIDA-IRP in accordance with NIH guidelines.

## Stereotaxic Surgery

All surgical procedures adhered to guidelines for aseptic technique. For electrode implantation, a drivable bundle of eight 25-um diameter formvar insulated nichrome wires (A-M systems, Carlsborg, WA) chronically implanted dorsal to VTA in the left or right hemisphere at 5.3 mm posterior to bregma, 0.7 mm laterally, and 7.5 mm ventral to the brain surface at an angle of 5° toward the midline from vertical. Some rats ($n = 9$) also received neurotoxic lesion of ipsilateral hippocampus by the infusion of NMDA (20 mg/ml) at seven sites in each hemisphere (see Fig. 1 for the surgical coordinates). Controls ($n = 5$) received sham lesions in which burr holes were drilled and the pipette tip lowered into the brain but no solution delivered. Cephalexin (15 mg/kg p.o.) was administered twice daily for two weeks post-operatively.

## Histology

All rats were perfused at the end of the experiment with phosphate-buffered saline (PBS) followed by 4% paraformaldehyde (Santa Cruz Biotechnology Inc., CA). Brains were cut in 40 μm sections and stained with thionin.

## Odor-guided choice task

Recording was conducted in aluminum chambers approximately 18" on each side with sloping walls narrowing to an area of 12" × 12" at the bottom. A central odor port was located above two fluid wells (Fig. 1c). Two lights were located above the panel. The odor port was connected to an air flow dilution olfactometer to allow the rapid delivery of olfactory cues. Odors were chosen from compounds obtained from International Flavors and Fragrances (New York, NY). Trials were signaled by illumination of the panel lights inside the box. When these lights were on, nosepoke into the odor port resulted in delivery of the odor cue to a small hemicylinder located behind this opening. One of three different odors was delivered to the port on each trial, in a pseudorandom order. At odor offset, the rat had 3 seconds to make a response at one of the two fluid wells. One odor instructed the rat to go to the left to get reward, a second odor instructed the rat to go to the right to get reward, and a third odor indicated that the rat could obtain reward at either well. Odors were presented in a pseudorandom sequence such that the free-choice odor was presented on 35% trials, and the left/right odors were presented in equal numbers. In addition, the same odor could be presented on no more than 3 consecutive trials. Once the rats were shaped to perform this basic task, we introduced blocks in which we independently manipulated the size of the reward or delay preceding reward delivery (Fig. 1d). For recording, one well was randomly designated as short and the other long at the start of the session (Fig. 1d, $1^{sh}$ and $1^{lo}$). In the second block of trials, these contingencies were switched (Fig. 1d, $2^{sh}$, $2^{lo}$). The length of the delay under long conditions followed an algorithm in which the side designated as long started off as 1 s and increased by 1 s every time that side was chosen until it became 3 s. if the rat continued to choose that side, the length of the delay increased by 1 s up to a maximum of 7 s. If the rat chose the side designated as long less than 8 out of the last 10 choice trials, then the delay was reduced by 1 s to a minimum of 3 s. The reward delay for long forced-choice trials was yoked to the delay in free-choice trials during these blocks. In the third and fourth blocks we held the delay preceding reward constant while manipulating the number of reward (Fig. 1d, $3^{bn}$, $3^{sm}$, $4^{bg}$ and $4^{sm}$). The switches between blocks were not explicitly signaled. The reward was a 0.05 ml bolus of 10% sucrose solution. The reward number used in delay blocks was the same as the reward used in the small reward blocks. For big reward, an additional bolus was delivered after gaps of 500 ms.

Rat training proceeded in three phases. Initially, rats underwent five sessions with only free-choice trials, where rewards in both wells were equal and constant. If a rat showed a strong side bias, one side was temporarily blocked to encourage balanced exploration. Next, rats experienced at least ten sessions with forced-choice trials only, using the same reward configuration as in recording sessions. This phase continued until rats achieved an 80% correct rate. Finally, free-choice and forced-choice trials were combined. The entire training lasted approximately 25 sessions and all rats could reliably complete all four trial blocks before the recording sessions.

## Single-unit recording

Wires were screened for activity daily; if no activity was detected, the rat was removed, and the electrode assembly was advanced 40 or 80 μm. Otherwise active wires were selected to be recorded, a session was conducted, and the electrode was advanced at the end of the session. Neural activity was recorded using Plexon OmniPlex Multi-channel Acquisition Processor systems (Dallas, TX). Signals from the electrode wires were amplified 20X by an op-amp headstage (Plexon Inc, HST/8o50-G20-GR), located on the electrode array. Immediately outside the training chamber, the signals were passed through a differential pre-amplifier (Plexon Inc, PBX2/16sp-r-G50/16fp-G50), where the single unit signals were amplified 50X and filtered at 150–9000 Hz. The single unit signals were then sent to the Multichannel Acquisition Processor box, where they were further filtered at 250–8000 Hz, digitized at 40 kHz and amplified at 1-32X. Waveforms ( > 2.5:1 signal-to-noise) were extracted from active channels and recorded to disk by an associated workstation

## Data analysis

Units were sorted using Offline Sorter software from Plexon Inc (Dallas, TX). Sorted files were then processed and analyzed in Neuroexplorer and Matlab (Natick, MA). Dopamine neurons were identified via a waveform analysis. Briefly cluster analysis was performed based on the half time of the spike duration and the ratio comparing the amplitude of the first positive and negative waveform segments. The center and variance of each cluster was computed without data from the neuron of interest, and then that neuron was assigned to a cluster if it was within 3 s.d. of the cluster's center. Neurons that met this criterion for more than one cluster were not classified. This process was repeated for each neuron. The putative dopamine neurons that showed an increase in firing to reward compared to baseline (400 ms before reward) were further classified as reward-responsive (t-test, $p < 0.05$). To analyze neural activity to reward, we examined the firing rate in the 400 ms beginning 100 ms after reward delivery.

## Computational models

We utilized a temporal-difference reinforcement learning (TDRL) algorithm within a partially-observable semi-Markov framework to simulate the evolution of reward prediction, and reward prediction errors, with experience on the behavioral task[32,34]. In this model, states $s$ are not observable (i.e., directly accessible to the behavioral agent), rather they are probabilistically associated with a set of observations corresponding to task events, such as the onset of the odor cue or the delivery of the reward. Each state is also probabilistically associated with a finite duration, $d$, that initiates at the time of the observation/s associated with a given state. We assumed that task events, modeled as non-empty observations, reliably signal transition to a new state, but that transitions may also occur with no corresponding event, i.e., an empty observation. Similar to other models in a partially observable setting, the conditional probabilities of each observation given a hidden state are specified by an observation function **O**, and the

probability of one state following another by a transition matrix, **T**. As rats were trained extensively on the odor-guided choice task, we assumed that they had learned a cognitive model for these aspects of the task, and therefore, function **O** and transition matrix **T** were known.

In this partially observable semi-Markov TDRL model, credit assignment requires estimating the current hidden state, which we model using an inference process that tracks the probability of having just transitioned out of state $s$ at time $t$, given the sequence of observations up to $t+1$, $\beta_{s,t}$[32]. As state transitions occur irregularly (as opposed to on every time point), performing this inference depends critically on an estimate of the likely dwell time $d$ of each state, captured by the dwell-time distribution, $D = P(d|s)$. To compute $\beta_{s,t}$, we rewrote it using Bayes' Rule:

$$\beta_{s,t} = P(s_t = s, \phi_t = 1 | o_1, \ldots, o_{t+1}) \quad (1)$$

$$= \frac{P(o_{t+1}|s_t = s, \phi_t = 1) * P(s_t = s, \phi_t = 1 | o_1, \ldots, o_{t+1})}{P(o_{t+1}|o_1, \ldots, o_t)}, \quad (2)$$

where the dummy variable, $\phi_t$, indicates whether a state transition happened at time $t$. According to the Markov property, the first term of the numerator in Eq. 2 is equal to the integration over state $s_t$, i.e., $P(o_{t+1}|s_t = s, \phi_t = 1) = \sum_{s' \in S} T_{s,s'} O_{s',o_{t+1}}$. Leveraging the assumption that $\beta_{s,t} = 1$ if $o_{t+1}$ is non-empty, the second term of the numerator, denoted $\alpha_{s,t}$, can be computed by integrating over all possible dwell times in state $s$, since the last non-empty observation:

$$\alpha_{s,t} = P(s_t = s, \phi_t = 1 | o_1, \ldots, o_t) \quad (3)$$

$$= \sum_{d=1}^{t_0} P(s_t = s, \phi_t = 1, d_t = d | o_1, \ldots, o_t) \quad (4)$$

$$= \sum_{d=1}^{t_0} \frac{P(o_{t-d+1}, \ldots, o_t | s_t = s, \phi_t = 1, d_t = d, o_1, \ldots, o_{t-d}) P(s_{t-d+1} = s, \phi_{t-d} = 1 | o_1, \ldots, o_{t-d})}{P(o_{t-d+1}, \ldots, o_t | o_1, \ldots, o_{t-d})} \quad (5)$$

$$= \sum_{d=1}^{t_0} \frac{O_{s,o_{t-d+1}} D_{s,d} P(s_{t-d+1} = s, \phi_{t-d} = 1 | o_1, \ldots, o_{t-d})}{P(o_{t-d+1}, \ldots, o_t | o_1, \ldots, o_{t-d})}, \quad (6)$$

where $t_0$ is the time that passed since the last non-empty observation. The last term of the numerator in Eq. 6 is the probability that the process left state $s$ at time $t - d$ weighted by the transition probability from $s_{t-d}$ to $s_{t-d+1}$, which is $\alpha_{s,t-d}$. Thus, $\alpha_{s,t}$ can be computed recursively,

$$P(s_{t-d+1} = s, \phi_{t-d} = 1 | o_1, \ldots, o_{t-d}) = \sum_{s' \in S} T_{s,s'} * P(s_{t-d} = s', \phi_{t-d} = 1 | o_1, \ldots, o_{t-d}) \quad (7)$$

$$= \sum_{s' \in S} T_{s,s'} * \alpha_{s',t-d}. \quad (8)$$

The denominator in Eq. 2, $P(o_{t+1}|o_1, \ldots, o_t)$, is computed by conditioning on state and $\phi_t$:

$$P(o_{t+1}|o_1, \ldots, o_t) = \sum_{\phi \in \{0,1\}} \sum_{s' \in S} P(o_{t+1}|o_1, \ldots, o_t, \phi_t = \phi, s_t = s') \\ * P(\phi_t = \phi, s_t = s' | o_1, \ldots, o_t) \quad (9)$$

$$= \sum_{s' \in S} P(o_{t+1}|\phi_t = 1, s = s') * \alpha_{s,t} + \sum_{s' \in S} P(o_{t+1}|s_{t+1} = s') \\ * \left( P(s_t = s | o_1, \ldots, o_t) - \alpha_{s,t} \right) \quad (10)$$

$$= \sum_{s' \in S} T_{s,s'} O_{s',o_{t+1}} * \alpha_{s,t} + \sum_{s' \in S} O_{s',o_{t+1}} * \left( P(s_t = s | o_1, \ldots, o_t) - \alpha_{s,t} \right), \quad (11)$$

where the belief $P(s_t = s | o_1, \ldots, o_t)$ is computed recursively as Eq. 6 by replacing $D_{s,d}$ with $P(d_t > d | s_t = s)$.

Vectorized prediction errors are gated by the probability of having just exited a state according to

$$\delta_{s,t} = \beta_{s,t} \left( e^{\tau E[d_{s,t}]} r_{t+1} + e^{\tau E[d_{s,t}]} E\left[\hat{V}_{s_{t+1}}\right] - \hat{V}_{s_t} \right). \quad (12)$$

Here, the future reward, $r_{t+1}$, and expected value of the successor state at time $t+1$, $E\left[\hat{V}_{s_{t+1}}\right]$ are each exponentially discounted by the expected dwell time spent in state $s$ before transition, $E[d_{s,t}]$. To regulate the strength of temporal discounting, we used a discount factor of $\tau = 0.05$. The expectation $E\left[\hat{V}_{s_{t+1}}\right]$ is computed by conditioning on the hypothesis that the process transitioned out of state $s$ at time $t$ and integrated over successor states $s'$:

$$E\left[\hat{V}_{s_{t+1}}\right] = \sum_{s' \in S} \hat{V}_{s'} * P(s_{t+1} = s' | s_t = s, \phi_t = 1, o_{t+1}) \quad (13)$$

$$= \sum_{s' \in S} \hat{V}_{s'} * \frac{T_{s,s'} O_{s',o_{t+1}}}{\sum_{s''} T_{s,s''} O_{s'',o_{t+1}}}. \quad (14)$$

The dwell time is unknown owing to the nature of partial observability, but it can be estimated by integrating over duration spent in state $s$ as for the computation of $\alpha$. The sum is taken out until the last non-empty observation is observed since the dwell time could not be longer.

$$E[d_{s,t}] = \sum_{d=1}^{t_0} d * P(d_t = d | s_t = s, \phi_t = 1, o_1, \ldots, o_{t+1}) \quad (15)$$

$$= \sum_{d=1}^{t_0} d * P(d_t = d | s_t = s, \phi_t = 1, o_1, \ldots, o_t) \quad (16)$$

$$= \frac{\sum_{d=1}^{t_0} d * P(d_t = d, s_t = s, \phi_t = 1, | o_1, \ldots, o_t)}{\alpha_{s,t}}. \quad (17)$$

Values for each state are updated with the total prediction error over all states, $\sum_{s \in S} \delta_{s,t}$, at each time point,

$$V_s \leftarrow V_s + \eta_r * E_{s,t} * \sum_{s \in S} \delta_{s,t}, \quad (18)$$

where $\eta_r = 0.5$ is the learning rate and controls the speed of learning the state value. The eligibility trace, $E_{s,t}$, records the visiting of each state from the start of a trial,

$$E_{s,t} = \max(\gamma * E_{s,t-1}, \beta_{s,t}), \quad (19)$$

where $\gamma = 0.95$ is a temporal decay parameter, determining how far the prediction error could backpropagate to the states preceeding the current state.

When a non-empty observation is observed, the dwell-time distribution is updated by a Gaussian density function that centers on the time that passed since the last non-empty observation.

$$\mathbf{D_s} \leftarrow \mathbf{D_s} + \eta_d * \beta_s * (K_d - D_s). \quad (20)$$

Here, $\eta_d = 0.3$ is the learning rate for these dwell-time distributions. $K_d$ is a Gaussian kernel with mean $d$, and standard deviation, $CV_d * d$, with coefficient of variation $CV_d = 0.05$. To keep $P(d|s)$ non vanishing for all reasonable dwell times, we fixed the baseline probability to $D_b = 10^{-4}$ for all $d$.

### Reward prediction error and neural firing

It has been suggested that dopamine neurons encode reward prediction error. We converted the total prediction error over all states, $\sum_{s \in S} \delta_s$, into equivalent firing rate and compared it with the averaged firing rate of dopamine neurons:

$$firing\ rate = \begin{cases} baseline + k1 * \sum_{s \in S} \delta_s, \sum_{s \in S} \delta_s \geq 0 \\ baseline + k2 * \sum_{s \in S} \delta_s, \sum_{s \in S} \delta_s < 0 \end{cases}. \quad (21)$$

Baseline firing was set to 3 Hz. $k1 = 5$ and $k2 = -2$ were the scale factors for positive and negative prediction errors, respectively. Different scale factors were used to reflect the fact that negative errors were underrepresented in vivo.

We did not formally fit the parameters to dopamine neurons' activities. Instead, we chose parameters manually to ensure that the reward prediction error signals quantitively match the neural activities. The results were robust and quantitively invariant across a wide range of parameter values. In addition, the results were not sensitive to the initial state value or dwell-time distribution. The simulations mimicked the task event timing used in the animal experiment. To reduce any potential variance caused by the initialization and inaccurate estimation of state values and dwell-time distributions, we excluded the first 10 sessions from the analysis shown in the results, and all results were averaged across 20 independent simulations.

For simplicity, we did not include free-choice trials but treated them similarly to forced-trials on the same side. We assumed that the model always selected the fluid well that led to the reward, since rats exhibited high accuracy (>80%) during the forced-choice trials. Fully modeling the free-choice trials and the corresponding choice did not change any of the reported results.

### Model 1

The task state space in model 1 was designed to reflect rats' physical location and observation when performing the task, composing seven different states, i.e., trial start, left cue, right cue, left well, right well, left reward 1, left reward 2, right reward 1, right reward 2 and inter-trial interval (Fig. 5a). The transition matrix, shown in the left panel of Fig. 5b, controlled the transitions between states. We included observations that rats indeed observed during the task, i.e., light onset, odor cues signaling the left and right choice, rewards, light offset, and a null (i.e., empty) observation (Fig. 5b, right panel). Each state brought about a non-empty observation that indicated a transition into that state. We assigned a high probability (0.95) to the only non-null observation and 0.05 to the null observation for each state. The remaining possible observations each was assigned a background probability of $10^{-4}$ to avoid the vanishing of probabilities, and the observation probabilities for each state were normalized by dividing their sum to ensure that the summation of observation probabilities for each state was 1.

We hypothesized that the HC-lesioned animals had difficulty in maintaining the hidden information after making choices, as what was observed in the OFC-lesioned animals. Therefore, we simulated an HC lesion by reducing the lesioned model's ability to retain information when choices were made. Specifically, we set the probabilities of

transitions from cue states to the well state on the same side to 0.55, and the probabilities of the transitions to the other well to 0.45 (Fig. 5b, left panel).

### Model 2

We constructed a hierarchical task space for model 2, with the lower-level describing the process happening in each trial and upper-level controlling the transition between blocks. The states in the lower-level were defined similarly to the states in model 1, except that separate states were introduced to describe the reward contingency in each block (Fig. 6a). The probabilities of transitions from the inter-trial state to each trial start states were updated with a learning rule,

$$T_{ITI, b_n} \leftarrow T_{ITI, b_n} + \eta_t * \left( \frac{T_{(o_1, \ldots, o_T | b_n)} * T_{ITI, b_n}}{\sum_{i=1}^{4} T_{(o_1, \ldots, o_T | b_i)} * T_{ITI, b_i}} - T_{ITI, b_n} \right), \quad (22)$$

where $T_{(o_1, \ldots, o_T | b_n)}$ was the probability of observing the sequence of observations, $o_1, \ldots, o_T$, in block n. The probability of transition from inter-trial interval to the block $n$ trial start state, $T_{ITI, b_n}$, would become larger if $o_1, \ldots, o_T$ were more likely to be observed in block n, and $\eta_t$ controlled the learning speed. Other transition probabilities were also not updated (i.e., those describing transitions between lower-level states within a trial) since they were not changed through training.

Task observations and their probabilities were assigned similarly to those in the model 1. Unlike state dwell-time distributions in model 1, which were updated to reflect the changing timing of the reward, the dwell-time distribution in the model 2 was fixed during the training process. This was because different temporal reward contingencies were represented by separate states in model 2. As a result, estimations of dwell time did not change, so we fixed dwell-time distributions for model 2 according to the mean delay to reward in the actual task for simiplify.

**HC lesion.** Hippocampus is widely recognized as an important brain region for representing cognitive states, which reflect an agent's location within a cognitive map. Therefore, we hypothesized that HCx rats failed to update and track their current block in a session. To simulate the effect of HC lesion, we introduced a higher variance between transitions in the top layer of the state space in our model (Fig. 6b). Specifically, $T_{ITI, b_n}$ was set to 0.15 whenever it was smaller than 0.15.

**OFC lesion.** As in the prior study, we assumed that the lesioned model could not differentiate between states following the chosen action and failed to accurately track the reward type associated with each well. As a result, the model ended up learning the same dwell-time distribution for both wells, as well as the transitions following the well states. To implement this, we blurred the probabilities of transitions from cue states to well states. Considering the dwell-time distribution and transition probabilities among low-level states in the hierarchical model were not plastic, we set an identical dwell-time distribution based on the average of two dwell-time distributions for reward delay for good states in each block before training. In addition, we allowed the reward 1 state to transit to either the reward 2 state or the intertrial interval state in blocks 3 and 4 for both wells (Fig. 7a).

### VS lesion

Consistent with the previous study, to simulate a full VS lesion in the model, we prevented the model from tracking the dwell time staying at each state and learning precise dwell time distributions for each state. We implemented this by reducing the amplitude of the dwell time, $E[d_{s,t}]$ (Eq. 15), and the Gaussian kernel, $K_d$ (Eq. 20), to zero (Fig. 7b). Consequently, the kernel carried zero probability mass at the current time and dwell-time distributions for each state remained at a baseline uncertainty for all finite time points. These manipulations blocked the formation of precise temporal expectations regarding the dwell time

of task states and impaired the model's capacity to infer unobservable transitions, such as the omission or delay of a reward.

## Reporting summary

Further information on research design is available in the Nature Portfolio Reporting Summary linked to this article.

## Data availability

The data generated in this study have been deposited in the Github and are accessible at https://github.com/YouGTakahashi/HCx_Delay_Task. Source data are provided with this paper.

## Code availability

All custom code used for reported analyzes in this study are available at https://github.com/YouGTakahashi/HCx_Delay_Task for the unit analyzes and https://github.com/zwzhang36/POSMDPs for the modeling.

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

## Acknowledgements

This work was supported by the Intramural Research Programs at the National Institute on Drug Abuse and the National Institute on Mental Health (Z1A-DA000587 to GS). The opinions expressed in this article are the authors' own and do not reflect the view of the NIH/DHHS. The authors have no conflicts of interest to report.

## Author contributions

Y.K.T., T.K., and G.S. conceived of and designed the experiments, Y.K.T. and M.M.C. conducted the behavioral training and single unit recording, and Z.Z. conducted the modeling after the experiment, with advice and assistance from A.J.L. All authors contributed to interpreting the data and writing the manuscript.

## Funding

## Competing interests

The authors declare no competing interests.
