## [Peer Review File · Nature Communications]

Expectancy-related changes in firing of dopamine neurons depend on hippocampusREVIEWER COMMENTS

Reviewer #1 (Remarks to the Author):

Takahashi et al. investigate the impact of unilateral hippocampal lesions on reward prediction error signalling in midbrain dopaminergic neurons. Compared to controls, HP-lesioned rats displayed normal cue-evoked prediction errors but lacked reward-evoked signals, suggesting the hippocampus plays a role in integrating block type information for reward prediction. A computational model explains both HP and previously reported OFC lesion effects.

The paper addresses an important and timely topic, is technically well done and reveals an interesting pattern of results. Pending replies to my concerns below I think it is suitable for Nature Communications.

Main Concerns:

1 The modelling is difficult to follow. The paper would benefit if the authors would provide specific information in the text about what the states are, what the observations are, which states have dwell time distributions and importantly, what the lesions do.

1A I don't have a good intuition for how not having block context can still leave cue-evoked expectations intact. Relatedly, I'm not sure I understand why in the lesion models there is no "model free" signal locked to the reward. If I understand correctly, in the OFC lesion model the rats have an intact association between door and well, such that they know which well to put their nose in when a specific odour comes. But once inside the well, they essentially forget which well they are currently in. In the HPC lesion model, the rats don't know which block they are in. They can smell a cue, retrieve its associated value, but when the cue is gone they forget what happened just now and can't really understand the reward?

1B It would be nice to revise figure 5 to show the transitions in the intact model and the lesioned model separately, and give a clear example about which states the model confuses when (same for Fig 6).

1C It would also be good to expand more on the behavioural predictions of the model. I expected that the value expectation from a previous block would carry over, especially for the Control group compared to the HCx group, suggesting an unexplored area in both behavior and cue-evoked responses. This generates predictions for both behavioral and cue-evoked responses which I do not think the authors have properly investigated.

2. I am concerned about whether the identification procedure of reward sensitive DA neurons has impacted results. Was neuron selection done after the lesions? If yes it seems problematic. I understand that the early vs late difference might be independent of the selection, but it's hard to be sure this is true. It also appears that after HPC lesions, DA neurons still have a small firing bump in response to rewards (which does not change across learning), but essentially no response to reward omissions. Is this a result, or just a reflection of the fact that DA neurons were defined based on having a difference between reward and no reward? Which rewards were taken in the identification procedure? Early rewards? Late rewards? It would be good to see a shuffling test that establishes independence of DA neuron selection and lesion effects.

3. How did the lesion impact reward- and cue-evoked DA signals in free choice trials? Figures 3-4 describe forced-choice trials only. This is interesting especially in light of the authors' previous finding whereby OFC degraded cue-evoked DA signals specifically in free-choice trials.

4. Could the authors help readers understand how the reported differences in DA firing do not lead to significant behavioral differences? Does the intact side counteract the effects of impaired DA firing? include analyzing free-choice trials, exploring block order effects, and considering simpler model alternatives.

5. The paper could provide more details in the paradigm. It's unclear if the well order was always Right-Left-Right-Left or randomized. Were the same odours used across blocks? This could have important implications of the behavioural and neural changes one would expect. One could hypothesize that cues would carry over value from previous blocks if motor and sensory overlap is given (see e.g. Shahar et al., 2019)

Minor:

- General note: the format of the submission with the figures at the end and the captions separated has made it harder to review the paper and follow the results.

- Have the authors considered a simpler 2 state model alternative to explain HPC lesions, that just has a 'good well' and 'bad well' representation? A corresponding analysis should consider block order. This model will make both behavioral and neural predictions.

- I think that in this experiment each block had 20 trials (p. 15 line 11) while in the previous experiment (2011 OFC study) each block had 40 trials. If this is the case it should be stated explicitly.

- Fig 1d, 2sh condition: why is there no red arrow in for the delayed reward, which in well 2 now is missing?

- Were different odours used between blocks? Also were block changes explicitly cued?

- In the discussion they ascribe OFC the unique role of being needed for hidden state inference – however in the current study block type is hidden as well – so one could say that HP has the same role but for longer time windows.

- It will be interesting to discuss this in light of the concept of event boundaries ascribed to the HP (Ben-Yakov & Henson, 2018, J Neuro; Hamamy et al, Nature Neuro, 2023)

- It would be interesting to discuss the possible interplay of HPC and OFC. Does HPC have a longer time window to integrate (non-observable) sequential task structure, which it provides to the OFC (see Schuck et al. 2019, Science)?

- Page 5 line 3 not “similar” rather “did not significantly differ”. Minor but important to make clear.

- Would be good to clarify how much time between blocks? can this be leveraged to look at stronger effect at animals or between blocks that had bigger time difference?

- The significance of the "first 8 trials" is unclear, Were forced choice trials included in the first 8 trials, potentially signaling a need to adjust for the previous block's influence? A linear model could consider the trial order's impact on this effect.

- Page 6 line 2: I'm not sure I understand why do you need both trial and early/late factors in the ANOVA

- Title of figure 2 caption is wrong (copied from another figure)

- Figure 7 left side is quite unclear: is C the cue-evoked or reward evoked responses? It would be beneficial to add some explanatory labels

Reviewer #2 (Remarks to the Author):

Takahashi et al., NComms 2024

The signaling of reward prediction errors by ventral tegmental area (VTA) dopamine (DA) neurons is one of the classic findings in neuroscience. This manuscript by Takahashi and colleagues investigates how the patterns of firing of VTA neurons is influenced by lesions to one part of the brain thought to be essential for cognitive maps, hippocampus. They report that in rat with lesions of hippocampus performing a rewarded task that DA neurons in the VTA do not exhibit the classic changes in firing to unexpected delivery of rewards as well as unexpected omission of rewards. Somewhat paradoxically, DA neurons firing rates in rats with hippocampus lesions do, however, change over the course of learning in a manner similar to what is seen in controls. To attempt to explain this seemingly contradictory pattern of results, the authors then develop a number of computational models to explain this pattern of effects, selectively “lesioning” different parts of the models to produce a pattern of model behavior that is similar to that shown by VTA neurons. These models indicate that what hippocampus was essential for in the firing patterns of DA neurons was a higher order model of the task.

This manuscript is really quite interesting and builds on a series of studies from this group looking at how different parts of the brain contribute to the basis of VTA DA prediction errors. The task is appropriate and using the same design as prior work is a major strength as it allows direct comparisons to that prior work. There are however some places where it needs some work, especially in the writing and framing of the question. Specifically, at present the motivation for the study is framed around the contrast of hippocampus to the role of orbitofrontal cortex in VTA prediction errors. This is fine but completely ignores the quite separate and extensive work on hippocampus in cognitive maps and the encoding of states which is one of the main results of the modeling. I also found the modeling work more than a little qualitative and thus not as convincing as it could be. These points are expanded on below with a number of other comments that the reviewers should consider.

1) The language used to frame the basis for the experiments in the introduction and throughout manuscript is a little jargony and not as precise as it could be. While this type of style can make the science more accessible and I'm all for reducing the stuffiness of science writing, but here it becomes slightly more unhelpful as it could be confusing to a reader, especially one not so steeped in the historical context of these experiments. For instance, on line 16 "Critically, these results were not consistent with our hypothesis going in that OFC provided information....". Here it is not clear what "going in" relates to. Similarly, a reader not so familiar with this area might wonder what a "cognitive mapping circuit" is related to. Another example, page 10, line 11: "...where a flat task space was use". What is a "flat task" space? I'm not trying to be difficult but I don't know what this is. The authors might want to have a non-specialist in this area look the paper over for places where language usage makes things unclear.

Related to the above, the introduction concentrates heavily on the comparison of this experiment looking at the effects of hippocampal lesions on DA neuron firing to those of OFC lesions. This is fine, but there is a substantial literature on the role of hippocampus in cognitive maps alone that really should be brought in here to frame the question (see work of Eichenbaum, Behrens, as well as the senior author of this paper, Geoff Schoenbaum). Indeed, lack of this point is a little puzzling as the best fitting model that is later developed is essentially a model that includes higher-level states or task contexts in order to provide a better fit of the observed data. Again, someone unfamiliar with this area of research would be left unaware of why such a model might be appropriate to develop given ideas about the hippocampus in cognitive maps. I would strongly recommend that the authors include more on the role of hippocampus in cognitive maps, states, etc in the introduction as well as throughout to frame the questions, results and modeling of the data to help to explain why they took the approach that they did.

The analysis of the effects of hippocampal lesions on DA neuron activity in response to rewards and omissions is reasonable as presented but doesn't really delve deeper into what has changed in the data. So, first what are the proportions of DA neurons in controls and hippocampal lesion animals that statistically exhibit positive and negative RPEs? Are these different between the groups? This point is potentially important as the analyses conducted so far only look at the population level distributions of differences in firing as opposed to how things are changed at the single neuron level. Second, the analyses presented in Figure 3F make it look like there might be a difference in the reward and omission responses in the last 5 trials of the blocks or potentially that responses to omissions are statistically different to baseline. Are either of these the case? Either way, to better show which difference are statistically significant in these plots (either between rewards/omissions or either condition versus baseline) the author should (where appropriate) include markers where responses to rewards and omissions and do statistically differ. Third, and I realize that this is a nit-picky point, the authors finish the analysis of hippocampus lesion effects on VTA RPE responses with a statement that "dopamine neurons recorded in rats with ipsilateral HC lesions failed to show normal bidirectional changes in firing...". I'm not sure that the analyses presented actually support such a strong statement. To make this claim you would need there to be effectively no modulation in any DA neuron. The distributions presented in figure 3 show that this isn't quite true. Some

neurons do show modulation (see outliers in Fig 3e). Fourth, I am interested to know if there is a relationship between the positive and negative RPEs at the single neuron level and if this changes as a result of hippocampus lesions. So, do VTA neurons that exhibit a robust change in firing rate in response to rewards also exhibit strong responses to omissions? A simple correlation at the single neuron level should show this. How does hippocampal lesions alter this (or not). This question is motivated by wondering if the loss of hippocampus also disrupts RPE signaling at the single neuron level, i.e. signaling of rewards and omissions at the single neuron level becomes uncoupled.

Quantitative assessments of model fits and development of models: On line 7 of page 9 it is reported that one of the models “did a good job of reproducing the pattern of reward prediction errors”. While I agree that the figures do match the responses of DA neurons, this statement is simply qualitative and feels quite thin. One person's good job can be another's dogs dinner. I may have missed it but I didn't get a good sense of how or whether the authors quantitatively tested how well their models fitted the data. This detail should be reported in the manuscript to help build confidence in the careful modeling work that has been done here. What would also be good to include to help convince a reader of the robustness/specificity of the models would be a little more information about model development. Here I'm interested to know how a model would behave if the higher-level block-like feature of the second model only had two states in it instead of four. Similarly if the model had more higher-level states in it than 4 how would it perform? The reason for this request is that at the moment the models rather just appear as perfectly fitting mirrors of the data without much idea of how they breakdown (or not).

In the section where the model is lesioned in different ways to reproduce the effects of OFC and ventral striatum (VS) lesions, the authors really need to provide more rationale for why they “lesioned” the models in the ways that they did to get the specific effects on VTA DA neuron firing. At the moment it is not really clear why say VS lesions could/should be recapitulated by “preventing the model to accurately learn the dwell time spent in each state”. Please include more information about the effects of both OFC and VS lesions on behavior and DA activity in order to frame/explain why these manipulations were done. Another way of approaching this could be to run through all of the different ways that the model can be altered/lesioned and show that when certain parts are altered that you get different outcomes and that this matches some of what has been seen in neural data. The point here is that there may be patterns of activity that do not match what was seen after OFC, VS or hippocampal lesions but might be predictive of future studies on the effects of lesions to other brain areas.

Minor:

Lesions: the authors report that lesions only destroyed ~50% of the hippocampus. Which parts of hippocampus were spared and could the lack of a complete lesion have impacted the results? A little more reporting of the spared areas in the results and discussion of the lack of complete lesion feels appropriate.

Page 6, line: typo “direct comparison OF the data”

Reviewer #3 (Remarks to the Author):

This study demonstrates how (by which computational mechanism) the hippocampus shapes dopamine prediction errors (PE).

Midbrain dopamine neurons were recorded from rats with damaged, or intact, hippocampus, while they performed a complex odor-guided reward seeking task. The task (the same as previously published by the same authors) is carefully designed to allow the recording of different types of PE (eg. cue-evoked, reward-upshift, or reward-downshift), reflecting the integration of different types of information by dopamine neurons.

The authors show that hippocampal lesioned animals present selective impairments in PE encoding: only reward-evoked activity was disrupted, but not cue-evoked activity.

To explain this surprisingly selective effect, the authors turned to computational modeling. They introduce a state-space model (a refinement of the author's previously published model) that assumes that animals/agents represent their current 'state' using a combination of external events (cues, rewards), memory of past event (response history), and knowledge of the high-level task structure (the "block" organization of the task).

They show that a synthetic lesion of part of the model representing the task structure reproduces the effect of hippocampal lesion on PE encoding.

This result is far from trivial (perhaps even counterintuitive) and speaks to the power of computational modeling of the task. The authors then briefly replace their previous results (about

OFC, or ventral striatum lesions in the same task) in the framework of this new computational model -- showing how each region appears to contribute differently to the encoding of PE by dopamine neurons.

--

Overall, the paper is excellent. The experimental design was carefully thought-out and expertly executed. The paper, while dealing with complex concepts, is very well written. My comments are minor.

1) Effect of hippocampal lesion on reward timing vs quantity.

Reward prediction errors were introduced by manipulating the delay to reward, or the number of rewards. Both manipulations were then combined in the result sections; Fig 1 shows positive PE trials, or negative PE trials without distinction about what caused this PE (delay to reward or reward number). That's totally fine, but I wonder if authors can clarify if hippocampal lesions produced similar deficits in both types of PE (reward timing vs quantity).

2) Presence in well as hidden state?

If I understand correctly, the state-space task model assumes that the animal's presence in the left or right well is a hidden state (cf. Fig 5b - right). In other words, if not for their working memory, an animal would not know in which well their snout is located (left vs right). However, I assume that some visual and/or proprioceptive cues might make this state not completely hidden. Perhaps the authors could clarify and discuss this point (is presence in port indeed considered a hidden state, and if so what are the limitations of this assumption).

3) Time of hippocampus involvement in PEs computations

This study shows what type of hippocampus-dependent information is accessible to dopamine neurons for PE encoding. I'm curious about the timeline of this. Do the authors envision this hippocampal contribution happening on the fly, as PEs are computed (if so via which pathway)? Or is it the case that the hippocampus allows for a type of learning (possibly consolidated in the neocortex) that can then be called upon to compute PEs?

I don't necessarily expect the authors to have an answer, but it's something that the authors might want to discuss briefly.

We would like to express our gratitude to the reviewers for their constructive feedback. We have considered all reviewer's comments and made the necessary revisions. As a result, we believe that these revisions have significantly enhanced the quality of our manuscript. We have provided a detailed point-to-point response to the reviewers' comments, with their comments in black and our responses in blue.

REVIEWER COMMENTS

Reviewer #1 (Remarks to the Author):

Takahashi et al. investigate the impact of unilateral hippocampal lesions on reward prediction error signalling in midbrain dopaminergic neurons. Compared to controls, HP-lesioned rats displayed normal cue-evoked prediction errors but lacked reward-evoked signals, suggesting the hippocampus plays a role in integrating block type information for reward prediction. A computational model explains both HP and previously reported OFC lesion effects.

The paper addresses an important and timely topic, is technically well done and reveals an interesting pattern of results. Pending replies to my concerns below I think it is suitable for Nature Communications.

Main Concerns:

1 The modelling is difficult to follow. The paper would benefit if the authors would provide specific information in the text about what the states are, what the observations are, which states have dwell time distributions and importantly, what the lesions do.

I am grateful to Reviewer 1 for pointing out the confusion. We included detailed information about the states, observations, and other information in the main text for the convenience of readers.

The changes were made as follows

line 26, page 12:

“This model’s state space was designed to be flat, simply reflecting rats’ physical locations and observations during the task, without any additional levels of organization related to how the trials were blocked or other factors. The state space consisted of seven states: trial start, left/right cue, left/right well, left/right rewards (1st and 2nd drop), and inter-trial interval, as illustrated in Fig. 5a. The transitions between states were governed by the transition matrix shown in Fig. 5b (left panel). Observations, i.e., light onset/off, odor cues

for left/right choices, rewards, and a null observation, indicated the transition to different states. The observation matrix showed the probability of each observation given each state (Fig. 5b, right panel).”

line 8, page 13:

“To model the effects of HC lesions, we blurred the ability of the model to maintain internal information about the transition probabilities between the odor cues and the correct well states. This was achieved by increasing the probabilities of transition from the left cue state to the right well states, and from the right cue state to the left well state, from baseline (10^{-4}) to 0.45 (pink color in Fig. 5a, 5b). This reflected the hypothesis that HC lesions would prevent the brain regions responsible for state estimation from maintaining the hidden information during the period after a response had been made, while the rats were waiting for reward, an effect, essentially like that caused by lesions of OFC in this task.”

line 9, page 15:

“Each cluster of states mimicked the states in the first model, but adapted to the unique reward contingencies of each block. The observation matrix from the first model was retained to control the probability of observation given states in each cluster (Fig. 5b).”

line 27, page 15:

“To implement this, we introduced larger uncertainty in the transition probabilities between the upper-level states by increasing the minimum value of the transition probabilities to each state to 0.15 (Fig. 6b).”

1A I don't have a good intuition for how not having block context can still leave cue-evoked expectations intact. Relatedly, I'm not sure I understand why in the lesion models there is no “model free” signal locked to the reward. If I understand correctly, in the OFC lesion model the rats have an intact association between door and well, such that they know which well to put their nose in when a specific odour comes. But once inside the well, they essentially forget which well they are currently in. In the HPC lesion model, the rats don't know which block they are in. They can smell a cue, retrieve it's associated value, but when the cue is gone they forget what happened just now and can't really understand the reward?

The RPEs differ when the rats estimate different states, even with the same observations and state values. The lesion model with a hierarchical task state space updates its estimation of the current block based on the reward feedback, although not as accurately as the intact model. Consequently, the cue-evoked RPE signal updates accordingly, leaving it intact.

We did not mean to suggest that the rats forget which well/block they are in. Instead, we propose that without a functioning OFC and HPC, the brain regions responsible for state

estimation are unable to use relevant information to estimate hidden states precisely, at least for calculating prediction errors. Consequently, the model (or the VTA dopamine neurons) would be uncertain about which well/block it was in, resulting in the RPE patterns that have been shown in the manuscript. The observed reward outcome in the fluid well strongly affected the state estimation in the lesioned model, leading to the predicted reward aligned with the observation and no prediction errors. We have added this in the discussion to help readers understand the assumptions behind it better. (line 21, page 20).

“Critically, the extensive training and the use of a unilateral lesion, which likely mitigates behavioral changes, allow us to explore the changes in DA firing caused by the removal of HC output without behavioral confounds.”

1B It would be nice to revise figure 5 to show the transitions in the intact model and the lesioned model separately, and give a clear example about which states the model confuses when (same for Fig 6).

The difference between the intact model and the lesioned model in the flat model (Fig. 5) was highlighted by the color-coded arrows. For the hierarchical model, we showed the differences in the Fig 6b. To enhance clarity, as recommended, we added two supplementary figures (supp Fig. 3 and supp Fig. 4) to provide an example of what states in the flat or the hierarchical model become confused in the lesioned model when an unexpected reward is delivered in the second block.

1C It would also be good to expand more on the behavioural predictions of the model. I expected that the value expectation from a previous block would carry over, especially for the Control group compared to the HCx group, suggesting an unexplored area in both behavior and cue-evoked responses. This generates predictions for both behavioral and cue-evoked responses which I do not think the authors have properly investigated.

Throughout the entire study, we used the same odors. Therefore, as reviewer 1 indicated, the value expectation from a previous block indeed carried over to the next, and the violation of the reward expectation induced RPEs. To clarify, we improved the description in the manuscript as follows: (line 22, page 4)

“These same three odors were consistently used throughout the entire study, with their corresponding associations with actions unchanged. To induce errors in reward prediction, we manipulated either the timing or the number of rewards delivered in each well across 4 blocks of trials (Fig. 1d). The switches between blocks were not explicitly signaled.”

We have also now tested a behavioral prediction based on the hierarchical model. Here is the prediction: According to the hierarchical model, receiving a large or delayed reward provides clear guidance on which side to choose in subsequent free-choice trials, as the other side will definitely be an immediate/small reward. However, receiving the immediate/small reward does not provide sufficient information since there are 2 blocks containing an immediate/small reward, so the alternative could either be a large or a delayed reward, making the decision more ambiguous.

To test this prediction, we used logistic regressions to analyze the effect of choices from the previous trial on the choice in subsequent free-choice trials. The model includes two variables: *choice* represents whether rats chose the high-valued or low-valued side in the previous trial, capturing any preference for repeating actions. *reward* indicates whether rats received an immediate/small reward or a long/big reward in the previous trial. The dependent variable, y , denotes whether rats choose the high-valued or low-valued side in the current free-choice trials. In the sessions when rats were proficient on the task, we found that both variables *choice* and *reward* significantly affected y , as shown in supp Fig. 5. Specifically, rats were more likely to choose the side with the high-valued reward after receiving a long/big reward compared to after receiving an immediate/small reward, which confirms the above prediction of our model. Notably this held true in both control and HCx rats; as noted below, we believe this reflects the fact that we only used ipsilateral lesions.

We did not explore behavioral predictions caused by the HCx because no significant behavioral changes were observed after HC lesion, presumably because of overtraining and the ipsilateral lesions that left one hemisphere intact to mediate normal behavior. This was done to avoid any confounding behavioral changes that could potentially affect the interpretation of neural changes. Therefore, we were unable to test the effects of hippocampal lesions on animal behavior, which was beyond the scope of the current manuscript and has been thoroughly investigated in other studies (Langston, Rosamund F., and Emma R. Wood. *Hippocampus* (2010); Wiltgen, Brian J., et al. *Journal of Neuroscience* (2006)). We believe our proposal for the contribution of the HC in the model is consistent with these reports.

2. I am concerned about whether the identification procedure of reward sensitive DA neurons has impacted results. Was neuron selection done after the lesions? If yes it seems problematic. I understand that the early vs late difference might be independent of the selection, but it's hard to be sure this is true. It also appears that after HPC lesions, DA neurons still have a small firing bump in response to rewards (which does not change across learning), but essentially no response to reward omissions. Is this a result, or just a reflection of the fact that DA neurons were defined based on having a difference between reward and no reward? Which rewards were taken in the identification procedure?

Early rewards? Late rewards? It would be good to see a shuffling test that establishes independence of DA neuron selection and lesion effects.

The neuron selection was done after the lesion. We extracted the response from the reward epoch from all the trials and compared them with the baseline (400 ms during the inter-trial interval before trial onset), not the response when the reward was omitted, and compared them with a t-test. If a unit showed a significantly higher firing at the reward epoch ($p < 0.05$), we categorized the unit as a “reward-responsive” neuron.

The selection procedure we used actually led to a strict criterion for our analysis. Those reward-responsive neurons showed little change in response to the reward compared to the period before the reward delivery, providing more convincing evidence supporting the dopamine neurons from the rats with ipsilateral lesions showed degraded bidirectional changes in response to rewards.

Therefore, considering the prevalence and features of these neurons being the same across groups, we do not believe that this selection procedure led to any of the differences we report. However we'd be happy to conduct any special shuffling procedures that the reviewer might suggest, if they are possible, on our data.

3. How did the lesion impact reward- and cue-evoked DA signals in free choice trials? Figures 3-4 describe forced-choice trials only. This is interesting especially in light of the authors' previous finding whereby OFC degraded cue-evoked DA signals specifically in free-choice trials.

We included both free-choice and forced-choice trials in our analysis of reward-evoked DA firing. We chose not to analyze them separately as this would significantly reduce the number of trials available for analysis, particularly since the most significant changes in reward-evoked firing typically occurred in the first trial following block switches.

However in response to this comment, we have separated them in an analysis presented in the supplemental. For the cue-evoked DA, we found, in the control rats, the firing of DA neurons in response to the free-choice cue reflected the more variable option (supp Fig. 2a), which is consistent with findings from Roesch et al., 2007. However, in the HCx group, the firing response to the free-choice cue in later trials was significantly lower than that to the high-valued cue (ANOVA, $p < 0.01$) and significantly higher than that to the low-valued cue (ANOVA, $p < 0.01$; supp Fig. 2b) Therefore, the firing of DA neurons to the free-choice cue in the HCx group did not indicate the 'better option' and appeared to reflect an 'intermediate' value between the two options.

We also expanded our model to include the free-choice odor. With the total expected value equal to the sum of action values weighted by the probability of choosing each

action (Eq. 1), the model with hierarchical task space successfully reproduced the pattern observed in both control and HCx rats (supp Fig. 2c, 2d).

$$value = value_{left} * p(left) + value_{right} * p(right), \quad (1)$$

$$p_{left} + p_{right} = 1. \quad (2)$$

Here is the intuition behind it. The control model accurately identifies the current block and chooses the action leading to a high-valued reward following the free-choice cue. Then, the expected value of the free-choice cue always equals that of the high-valued cue in each block. In contrast, the lesion model lacks a clear estimation of the current block, leading to a lower probability of choosing the high-valued side, reducing the overall expected value. (Please note that this model only simulates dopamine neurons in the hemisphere where HX is lesioned. It does not assume that HCx rats choose the high-valued side with a lower probability.) Consequently, RPEs evoked by the free-choice odor are smaller than the high-valued cue in the lesion model.

The cue-evoked DA response in free choice trials and model simulation are now included in the manuscript, and related figures and details can be found in the supplementary information.

line 24, page 10.

“On free-choice trials, the firing during the presentation of the free-choice cue reflected the more variable option (supp Fig. 2a).”

Line 6, page 11.

“The only exception to this was in activity to the free-choice cue in later trials, which was significantly lower than that to the high-valued cue (ANOVA, $p < 0.01$) and significantly higher than that to the low-valued cue (ANOVA, $p < 0.01$; supp Fig. 2b).”

line 15, page 16.

“The dopamine responses in free-choice trials were also captured by this model with hierarchical task space (supp Fig. 1c, 1d; detailed methods and explanation could be found in the supplementary information).”

4. Could the authors help readers understand how the reported differences in DA firing do not lead to significant behavioral differences? Does the intact side counteract the effects of impaired DA firing? include analyzing free-choice trials, exploring block order effects, and considering simpler model alternatives.

Yes, it is indeed the case. The overtraining and intact side presumably mitigate any behavioral effects caused by the impaired side, such as those reported in the studies cited above. Therefore, we can explore the DA firing changes induced by the ipsilateral

lesion of HC without the influences from the behavioral changes. We have added to the discussion to try to clarify that for readers (line 21, page 20).

“Critically, the extensive training and the use of a unilateral lesion, which likely mitigates behavioral changes, allow us to explore the changes in DA firing caused by the removal of HC output without behavioral confounds. ”

5. The paper could provide more details in the paradigm. It's unclear if the well order was always Right-Left-Right-Left or randomized. Were the same odours used across blocks? This could have important implications of the behavioural and neural changes one would expect. One could hypothesize that cues would carry over value from previous blocks if motor and sensory overlap is given (see e.g. Shahar et al., 2019)

Sorry for the confusion. We used the same odors throughout the study. The cues did carry over values from previous blocks. Therefore, RPEs were introduced when we changed the reward outcome in the subsequent block. The block order was not fully randomized. However, the starting side was randomized, meaning either the left or the right well could be associated with an immediate reward in the first block. The delay condition was always first, as it is more challenging for the rats. If it is left for second, it often is not completed. Similar effects for both delay and size conditions were observed, which was demonstrated in supp Fig. 1.

To clarify, we have added this on line 22, page 4:

“These same three odors were consistently used throughout the entire study, with their corresponding associations with actions unchanged .”

line 30, page 4:

“Either well 1 or well 2 in Fig. 1d could be on the left side, and the other was on the right side, which was counterbalanced between sessions.”

Minor:

- General note: the format of the submission with the figures at the end and the captions separated has made it harder to review the paper and follow the results.

Sorry for the inconvenience, we have placed the figures in the main text at the points where they are referenced.

- Have the authors considered a simpler 2 state model alternative to explain HPC lesions, that just has a 'good well' and 'bad well' representation? A corresponding analysis should

consider block order. This model will make both behavioral and neural predictions.

Yes, we did explore a simpler two-state model. However, it had the same issue as the flat model. Particularly, when we represented wells with 'good well' and 'bad well' states, these states would learn the reward outcomes in two wells by the end of the 2nd block, which was long-delayed rewards and short-delayed rewards. However, these reward outcomes did not match the large reward condition in the left well during the 3rd block. Consequently, this model predicted a positive RPE upon the delivery of the 2nd drop of water, which was inconsistent with what we observed. Therefore, we did not include this model in the manuscript.

- I think that in this experiment each block had 20 trials (p. 15 line 11) while in the previous experiment (2011 OFC study) each block had 40 trials. If this is the case it should be stated explicitly.

We used the same setting in both studies. The blocks switched after approximately 60 trials. This randomness prevented the rats from predicting when a block switch would occur. Additionally, switches did not take place if rats had chosen the high-value side less than 60% in the last 10 free-choice trials, further preventing rats from anticipating block switches. Notably, during recordings, the correct rate in free-choice trials exceeded 80% before the block switches, as depicted in Fig. 1e.

On line 11, page 15, we meant that the proportion of free-choice trials was 7 out of 20.

To clarify, we have changed this on line 11, page 25:

“Odors were presented in a pseudorandom sequence such that the free-choice odor was presented on 35% trials, and the left/right odors were presented in equal numbers .”

- Fig 1d, 2sh condition: why is there no red arrow in for the delayed reward, which in well 2 now is missing?

The red arrow and blue arrow in Fig. 1d indicate where the negative and positive RPEs were introduced, respectively. In the 2sh condition, the short delayed reward induced a positive reward prediction error, leading rats to perceive the reward as being delivered earlier than anticipated. Consequently, they did not expect the longer-delayed reward from the first block, resulting in no negative RPE, which has been investigated in Fiorillo, Christopher D. et al. Nature Neuroscience (2008) and Takahashi, Yuji K., et al. Nature Neuroscience (2023).

- Were different odours used between blocks? Also were block changes explicitly cued?

The same odors were used between blocks, and block switches were not explicitly cued. We clarified it by adding an explanation at line 22, page 4:

“These same three odors were consistently used throughout the entire study, with their corresponding associations with actions unchanged. To induce errors in reward prediction, we manipulated either the timing or the number of rewards delivered in each well across 4 blocks of trials (Fig. 1d). The switches between blocks were not explicitly signaled.”

- In the discussion they ascribe OFC the unique role of being needed for hidden state inference – however in the current study block type is hidden as well – so one could say that HP has the same role but for longer time windows.

We agree with the reviewer’s observation. At the functional level, both OFC and HP are important for tracking or representing the hidden information with HP particularly important for longer time duration. At an implement level, the HP lesions blurred the transitions between high-level states, while OFC lesions blurred the transitions between low-level states.

- It will be interesting to discuss this in light of the concept of event boundaries ascribed to the HP (Ben-Yakov & Henson, 2018, J Neuro; Hamamy et al, Nature Neuro, 2023)

- It would be interesting to discuss the possible interplay of HPC and OFC. Does HPC have a longer time window to integrate (non-observable) sequential task structure, which it provides to the OFC (see Schuck et al. 2019, Science)?

These two points are interesting and have been discussed in the Discussion section as follows:

Line 2, page 22

“Specifically, updating the estimation of the trial block requires integrating information about odor identity and reward outcome across preceding trials. This process could be facilitated by the reactivation of HC neurons in patterns similar to those experienced during previous trials, which has been widely described in many previous studies. This reactivation could replay the information over a long time window and be triggered by a salient stimulus or event boundary that marks the end of one series of events and the beginning of another, such as rewards and intertrial interval. Without an intact HC and the reactivation of HC neurons, the estimation of the block may become more uncertain.”

Line 21, page 22

“Although we have shown the distinct contributions of the OFC and HC to the cognitive map, how they interact and contribute collectively remains a mystery. Our findings shed

light on this complex interaction. It appears that the estimation of the current block or upper-level state, potentially performed by HC neurons during the intertrial interval, depends on the lower-level states within a single trial, which might be estimated online by the OFC neurons. In turn, the estimation of the current upper-level state aids in the estimation of lower-level states. Thus, HC might utilize the lower-level states estimated by OFC to estimate the upper-level states and send this information back to OFC for further lower-level state estimation.”

- Page 5 line 3 not “similar” rather “did not significantly differ”. Minor but important to make clear.

Thanks for the suggestions. We have made changes accordingly.

- Would be good to clarify how much time between blocks? can this be leveraged to look at stronger effect at animals or between blocks that had bigger time difference

As previously stated, block switches occurred after approximately 60 trials, with switches only happening if rats had chosen the high-value side more than 60% in the last 10 free-choice trials. Slight randomness was introduced in the trial number in each block to prevent the rats from predicting when a block switch would occur. Given extensive training, all rats reached a high accuracy within 60 trials after switches, resulting in low variance in block length and the time between blocks. No additional delay existed between blocks. Therefore, it is hard for us to conduct the analysis suggested by the reviewer.

- The significance of the "first 8 trials" is unclear, Were forced choice trials included in the first 8 trials, potentially signaling a need to adjust for the previous block's influence? A linear model could consider the trial order's impact on this effect.

Only free-choice trials were included in Fig. 1e, which indeed showed that rats adjusted their behavior for the previous block's influence. The choices in the force-choice trials were also affected by the block switches, which were shown in Fig. 1f and 1e.

- Page 6 line 2: I'm not sure I understand why do you need both trial and early/late factors in the ANOVA

The factor 'trial' indicated the trial number. By using 'early/late', we can determine if there were changes in the response between the early or late phase of a block. By including the trial number, we can test if there was a gradual change in the response across trials.

- Title of figure 2 caption is wrong (copied from another figure)

Sorry for the oversight. We have corrected it now.

- Figure 7 left side is quite unclear: is C the cue-evoked or reward evoked responses? It would be beneficial to add some explanatory labels

Fig.7c described the reward-evoked response, and we added more explanatory labels to make it clearer.

Reviewer #2 (Remarks to the Author):

Takahashi et al., NComms 2024

The signaling of reward prediction errors by ventral tegmental area (VTA) dopamine (DA) neurons is one of the classic findings in neuroscience. This manuscript by Takahashi and colleagues investigates how the patterns of firing of VTA neurons is influenced by lesions to one part of the brain thought to be essential for cognitive maps, hippocampus. They report that in rat with lesions of hippocampus performing a rewarded task that DA neurons in the VTA do not exhibit the classic changes in firing to unexpected delivery of rewards as well as unexpected omission of rewards. Somewhat paradoxically, DA neurons firing rates in rats with hippocampus lesions do, however, change over the course of learning in a manner similar to what is seen in controls. To attempt to explain this seemingly contradictory pattern of results, the authors then develop a number of computational models to explain this pattern of effects, selectively “lesioning” different parts of the models to produce a pattern of model behavior that is similar to that shown by VTA neurons. These models indicate that what hippocampus was essential for in the firing patterns of DA neurons was a higher order model of the task.

This manuscript is really quite interesting and builds on a series of studies from this group looking at how different parts of the brain contribute to the basis of VTA DA prediction errors. The task is appropriate and using the same design as prior work is a major strength as it allows direct comparisons to that prior work. There are however some places where it needs some work, especially in the writing and framing of the question. Specifically, at present the motivation for the study is framed around the contrast of hippocampus to the role of orbitofrontal cortex in VTA prediction errors. This is fine but completely ignores the quite separate and extensive work on hippocampus in cognitive maps and the encoding of states which is one of the main results of the modeling. I also found the modeling work more than a little qualitative and thus not as convincing as it could be. These points are expanded on below with a number of other comments that the reviewers should consider.

1) The language used to frame the basis for the experiments in the introduction and throughout manuscript is a little jargony and not as precise as it could be. While this type of style can make the science more accessible and I'm all for reducing the stuffiness of science writing, but here it becomes slightly more unhelpful as it could be confusing to a reader, especially one not so steeped in the historical context of these experiments. For instance, on line 16 "Critically, these results were not consistent with our hypothesis going in that OFC provided information....". Here is it not clear what "going in" relates to. Similarly, a reader not so familiar with this area might wonder what a "cognitive mapping circuit" is related to. Another example, page 10, line 11: "...where a flat task space was use". What is a "flat task" space? I'm not trying to be difficult but I don't know what this is. The authors might want to have a non-specialist in this area look the paper over for places where language usage makes things unclear.

We thank the reviewer for pointing this out. We have clarified the jargon at its first use or have avoided using it. The changes are as follows:

Replace "cognitive mapping circuit": "Notably, these data led to the hypothesis that the OFC is critical for representing the current state in the cognitive map, which is relied on by other brain regions, like midbrain dopamine neurons." (line 20, page 3)

Explain "flat task space": "This model's state space was designed to be flat, simply reflecting rats' physical locations and observations during the task, without any additional levels of organization related to how the trials were blocked or other factors." (line 26, page 12)

Related to the above, the introduction concentrates heavily on the comparison of this experiment looking at the effects of hippocampal lesions on DA neuron firing to those of OFC lesions. This is fine, but there is a substantial literature on the role of hippocampus in cognitive maps alone that really should be brought in here to frame the question (see work of Eichenbaum, Behrens, as well as the senior author of this paper, Geoff Schoenbaum). Indeed, lack of this point is a little puzzling as the best fitting model that is later developed is essentially a model that includes higher-level states or task contexts in order to provide a better fit of the observed data. Again, someone unfamiliar with this area of research would be left unaware of why such a model might be appropriate to develop given ideas about the hippocampus in cognitive maps. I would strongly recommend that the authors include more on the role of hippocampus in cognitive maps, states, etc in the introduction as well as throughout to frame the questions, results and modeling of the data to help to explain why they took the approach that they did.

We appreciate the suggestions. In the Introduction, we have expanded upon the role of the HC in the cognitive map, in addition to the OFC, as follows (line 23, page 3):

"HC is another critical brain region implicated in encoding the task state, however the distinct contributions of the HC and the OFC to the cognitive map remain unclear. Early

research indicates that HC neurons encode animals' current location and guide goal-directed navigation. HC is also crucial for episodic memory, and HC neuron activity differs across various contexts, reflecting the situation or environment in which a series of events occur. More recent studies suggest that the HC is involved in organizing memories within context. The replay or reactivation of HC neurons integrates information essential for understanding the current context, which can be triggered by behavioral events, such as the completion of a trial or receiving a reward."

The analysis of the effects of hippocampal lesions on DA neuron activity in response to rewards and omissions is reasonable as presented but doesn't really delve deeper into what has changed in the data. So, first what are the proportions of DA neurons in controls and hippocampal lesion animals that statistically exhibit positive and negative RPEs? Are these different between the groups? This point is potentially important as the analyses conducted so far only look at the population level distributions of differences in firing as opposed to how things are changed at the single neuron level.

We found that 13 and 7 out of 44 units exhibited significant changes in response to unexpected reward delivery and omission, respectively, in the sham rats. In the control rats, 5 and 2 out of 66 units showed significant changes in response to unexpected reward delivery and omission, respectively. Chi-square tests were applied and indicated more neurons in sham rats showing significant changes in response to unexpected reward delivery ($p=0.0023$) and omission ($p=0.016$), compared to the HCx rats.

Line 14, page 8

"To quantify these changes, we compared firing in the first five and last five trials with two-tailed t-tests and found 13 and 7 out of 44 neurons showing significant changes for the positive and negative prediction errors, respectively."

Line 2, page 9

"Only 5 and 2 out of 66 neurons showed significant changes for the positive and negative prediction errors, respectively, numbers which were significantly less than what was found in the sham rats (Chi-square test, $p=0.0023$ and 0.016 for positive and negative prediction errors, respectively)."

Second, the analyses presented in Figure 3F make it look like there might be a difference in the reward and omission responses in the last 5 trials of the blocks or potentially that responses to omissions are statistically different to baseline. Are either of these the case? Either way, to better show which difference are statistically significant in these plots (either between rewards/omissions or either condition versus baseline) the author should (where

appropriate) include markers where responses to rewards and omissions and do statistically differ.

We compared the DA response evoked by the expected reward delivery and omission using two-tailed t-tests and Bonferroni correction, and found no significant difference in each of the last 5 trials of blocks in both sham and HCx rats. Since we did not intend to compare the firing induced by the positive prediction errors directly to that induced by negative prediction errors, we chose not to include this specific comparison in our manuscript but to provide the analysis results here.

		-T5	-T4	-T3	-T2	-T1
Ctrl	T value	-0.582	-0.601	0.287	0.420	0.399
	P value	0.564	0.551	0.776	0.677	0.692
HCx	T value	0.866	1.391	1.714	1.133	2.463
	P value	0.390	0.169	0.091	0.261	0.016

Table 1. Compare firing induced by positive and negative prediction error with paired t-tests.

This comment did prompt us to go back and redo our statistics to directly compare reward and omission responses with the baseline level using ANOVA analyses. We observed main effects relative to the baseline and interactions with early/late phases for both PPE and NPE in control subjects, but not in HCx-lesioned rats (Table. 2, 3). The sole exception was the interaction for PPE (Table. 3), which was also weaker than that observed in the control rats. Since the results presented here are consistent with what we found using three-way ANOVAs in the manuscript, which provide more comprehensive information, we choose to use the original ANOVA format in the manuscript.

CONTROL					
2-way ANOVA (REWARD(PPE/Baseline) x Early/Late)					
EFFECT	Wilks Value	F	Effect df	Error df	p
REWARD	0.688	3.532	5	39	.010*
EARLY/LA	0.606	5.074	5	39	.001*
REWARD*EARLY/LA	0.517	7.284	5	39	.000*
2-way ANOVA (REWARD(NPE/Baseline) x Early/Late)					
EFFECT	Wilks Value	F	Effect df	Error df	p
REWARD	0.87	1.17	5	39	0.341
EARLY/LA	0.75	2.595	5	39	.040*
REWARD*EARLY/LA	0.674	3.771	5	39	.007*

Table 2. Comparison of positive prediction error (PPE) and negative prediction error (NPE) with baseline using two-way ANOVA in control rats.

HCx					
2-way ANOVA (REWARD(PPE/Baseline) x Early/Late)					
EFFECT	Wilks Value	F	Effect df	Error df	p
REWARD	0.873	1.768	5	61	0.133
EARLY/LA	0.946	0.698	5	61	0.627
REWARD*EARLY/LA	0.744	4.205	5	61	.002*
2-way ANOVA (REWARD(NPE/Baseline) x Early/Late)					
EFFECT	Wilks Value	F	Effect df	Error df	p
REWARD	0.891	1.490	5	61	0.206
EARLY/LA	0.918	1.087	5	61	0.377
REWARD*EARLY/LA	0.888	1.536	5	61	0.192

Table 3. Comparison of positive prediction error (PPE) and negative prediction error (NPE) with baseline using two-way ANOVA in control rats.

Third, and I realize that this is a nit-picky point, the authors finish the analysis of hippocampus lesion effects on VTA RPE responses with a statement that “dopamine neurons recorded in rats with ipsilateral HC lesions failed to show normal bidirectional changes in firing...”. I’m not sure that the analyses presented actually support such a strong statement. To make this claim you would need there to be effectively no modulation in any DA neuron. The distributions presented in figure 3 show that this isn’t quite true. Some neurons do show modulation (see outliers in Fig 3e).

We are sorry for the previously assertive tone. We have revised our language to represent the findings supported by our data more accurately.

Line 17, page 9

Change to “Thus dopamine neurons recorded in rats with ipsilateral HC lesions show degraded bidirectional changes in firing – presumed to be reward prediction errors - in response to manipulations of reward.”

Fourth, I am interested to know if there is a relationship between the positive and negative RPEs at the single neuron level and if this changes as a result of hippocampus lesions. So, do VTA neurons that exhibit a robust change in firing rate in response to rewards also exhibit strong responses to omissions? A simple correlation at the single neuron level should show this. How does hippocampal lesions alter this (or not). This question is motivated by wondering if the loss of hippocampus also disrupts RPE signaling at the single neuron level, i.e. signaling of rewards and omissions at the single neuron level becomes uncoupled.

As the reviewer suggested, we subtracted the average normalized firing in late trials from that in early trials and plotted the changes in firing in response to reward delivery and

omission. A significant correlation between these index scores was found in the sham rats but not in the HCx rats, which suggested that the HC lesion disrupts RPE signaling at the single neuron level.

Figure 1. The relationship between the positive and negative prediction errors at the single neuron level in the control and HCx rats. (a) The change in normalized firing, which is the average normalized firing in late trials subtracted from that in early trials, in response to reward delivery correlates with that in response to reward omission in control rats. (b) This correlation was not found in HCx rats.

Quantitative assessments of model fits and development of models: On line 7 of page 9 it is reported that one of the models “did a good job of reproducing the pattern of reward prediction errors”. While I agree that the figures do match the responses of DA neurons, this statement is simply qualitative and feels quite thin. One person's good job can be another's dogs dinner. I may have missed it but I didn't get a good sense of how or whether the authors quantitatively tested how well their models fitted the data. This detail should be reported in the manuscript to help build confidence in the careful modeling work that has been done here. What would also be good to include to help convince a reader of the robustness/specificity of the models would be a little more information about model development. Here I'm interested to know how a model would behave if the higher-level block-like feature of the second model only had two states in it instead of four. Similarly if the model had more higher-level states in it than 4 how would it perform? The reason for this request is that at the moment the models rather just appear as perfectly fitting mirrors of the data without much idea of how they breakdown (or not).

In the manuscript, we presented qualitative evidence to demonstrate that the model with hierarchical task space offers a better explanation of our findings, which we felt was

convincing. However in response to this comment and to rule out the possibility that this was due to arbitrarily selected parameters and to further substantiate the robustness of our simulation, we have added a formal model fitting procedure with multiple starting points. Due to the complexity of running each simulation, we limited our fitting to three parameters that we believe most significantly affect the model results. For the flat model, these parameters are the learning rate for value, the dwell distribution, and the transition probabilities from cue states to the well state on the same side in the lesion model. For the hierarchical model, we fitted the learning rate for value, the transition probability, and the transition uncertainty in the lesion model. Then, we calculated and compared the likelihood of observing the DA response pattern in rats with flat and hierarchical structures. A significantly higher likelihood was found in the hierarchical model, suggesting it better explains the DA response.

We now include the model fitting and comparison in our manuscript, line 18, page 16.

“To rule out the possibility that the distinct pattern of prediction errors between two models was from arbitrarily chosen parameters, we fitted models by maximizing the likelihood of observing dopamine neurons firing in vivo. Due to the simulation's complexity, we fitted three key free parameters in each model that we believe most significantly affect the prediction error. For the flat model, we adjusted the learning rate for value, the dwell distribution, and the transition probabilities from cue states to the well state on the same side in the lesion model. For the hierarchical model, we adjusted the learning rate for value, the transition probability, and the transition uncertainty in the lesion model. With the best-fitting parameters, we calculated the likelihoods for two models and compared them using a likelihood ratio test. A significantly higher likelihood ($p=9.6e-31$, Chi-square test) was found in the second model with hierarchical task space, demonstrating it provides a more accurate explanation of the activity of dopamine neurons.”

The hierarchical model has four high-level states because we have four different blocks in our behavioral task. Each high-level state corresponds to one of four blocks in our task. Thus, simulating the task with only 2 or more than 4 high-level states is inappropriate.

In the section where the model is lesioned in different ways to reproduce the effects of OFC and ventral striatum (VS) lesions, the authors really need to provide more rationale for why they “lesioned” the models in the ways that they did to get the specific effects on VTA DA neuron firing. At the moment it is not really clear why say VS lesions could/should be recapitulated by “preventing the model to accurately learn the dwell time spent in each state”. Please include more information about the effects of both OFC and VS lesions on behavior and DA activity in order to frame/explain why these manipulations were done. Another way of approaching this could be to run through all of the different ways that the

model can be altered/lesioned and show that when certain parts are altered that you get different outcomes and that this matches some of what has been seen in neural data. The point here is that there may be patterns of activity that do not match what was seen after OFC, VS or hippocampal lesions but might be predictive of future studies on the effects of lesions to other brain areas.

Thank you for the suggestions. In our previous study, we described the effects of OFC and VS lesions on DA activity and the motivation behind the modeling. To enhance the clarity of our manuscript and assist readers who may not be familiar with our previous work, we have included a brief summary of this information. The changes have been made on Line 13, Page 18 as follows:

“Dopamine neurons in rats with ipsilateral OFC lesions failed to suppress firing to reward omission and exhibited weaker but also more persistent increases in firing to unexpected rewards (citation). A thorough model comparison showed that the ambiguity between left and right well states hindered learning and resulted in weaker but more constant prediction errors, providing the best explanation for the experimental findings. To reproduce this effect, we modeled the effects of OFC lesions in the hierarchical model by blurring the model's ability to maintain internal information within each block, specifically affecting the transition after actions (Fig. 7a), which caused the model to be unable to differentiate the two wells or track the reward associated with each well. The resultant lesioned model reproduced the altered prediction errors in response to rewards observed in the prior study in rats with OFC lesions (Fig. 7c).”

Line 23, Page 18

“Dopamine neurons in rats with ipsilateral VS lesions exhibited intact prediction errors in response to changes in the number of rewards but showed no error signals in response to changes in reward timing on the order of several seconds (citation), suggesting the normal temporal expectation was deficient due to the VS lesion. To reproduce this effect, we modeled the effects of VS lesions by preventing the model from accurately learning the dwell time spent in each state (Fig. 7b), which caused it to be unable to form precise temporal expectations or deduce unobservable state transitions. Consequently, there were no prediction errors when the reward timing changed or the reward was omitted but normal errors when additional rewards were delivered (Fig. 7d), consistent with the response in rats with VS lesions.”

Minor:

Lesions: the authors report that lesions only destroyed ~50% of the hippocampus. Which parts of hippocampus were spared and could the lack of a complete lesion have impacted the results? A little more reporting of the spared areas in the results and discussion of the lack of complete lesion feels appropriate.

We divided the hippocampus into dorsal and ventral parts and recalculated the lesioned area in these two subregions. The lesions affected 60% (55-66%) of the dorsal and 51% (37-58%) of the ventral parts. Our lesions resulted in more damage in the dorsal than in the ventral subregions. However, given the small number of animals, it is challenging to analyze correlations between the extent of lesions in each subregion and the changes in dopamine response. We are open to conducting further investigations if the reviewer has more specific suggestions.

Page 6, line: typo “direct comparison OF the data”

Corrected

Reviewer #3 (Remarks to the Author):

This study demonstrates how (by which computational mechanism) the hippocampus shapes dopamine prediction errors (PE).

Midbrain dopamine neurons were recorded from rats with damaged, or intact, hippocampus, while they performed a complex odor-guided reward seeking task. The task (the same as previously published by the same authors) is carefully designed to allow the recording of different types of PE (eg. cue-evoked, reward-upshift, or reward-downshift), reflecting the integration of different types of information by dopamine neurons.

The authors show that hippocampal lesioned animals present selective impairments in PE encoding: only reward-evoked activity was disrupted, but not cue-evoked activity.

To explain this surprisingly selective effect, the authors turned to computational modeling. They introduce a state-space model (a refinement of the author's previously published model) that assumes that animals/agents represent their current 'state' using a combination of external events (cues, rewards), memory of past event (response history), and knowledge of the high-level task structure (the "block" organization of the task).

They show that a synthetic lesion of part of the model representing the task structure reproduces the effect of hippocampal lesion on PE encoding.

This result is far from trivial (perhaps even counterintuitive) and speaks to the power of computational modeling of the task. The authors then briefly replace their previous results (about OFC, or ventral striatum lesions in the same task) in the framework of this new computational model -- showing how each region appears to contribute differently to the encoding of PE by dopamine neurons.

Overall, the paper is excellent. The experimental design was carefully thought-out and expertly executed. The paper, while dealing with complex concepts, is very well written. My comments are minor.

Thanks to Reviewer 3 for the kind words and appreciation.

1) Effect of hippocampal lesion on reward timing vs quantity.

Reward prediction errors were introduced by manipulating the delay to reward, or the number of rewards. Both manipulations were then combined in the result sections; Fig 1 shows positive PE trials, or negative PE trials without distinction about what caused this PE (delay to reward or reward number). That's totally fine, but I wonder if authors can clarify if hippocampal lesions produced similar deficits in both types of PE (reward timing vs quantity).

As suggested, we analyzed the dopamine responses for each condition separately and found a significant effect of HCx on the dopamine response caused by both the delay and size switches. Furthermore, no significant differences were observed in the dopamine changes caused by the delay and size switches in either sham or HCx rats. We have included a figure in the supplementary (supp Fig. 1) material to illustrate these findings and incorporated this information into the main text.

line 22 page 8:

“These observations were consistent when responses induced by changes in timing or the number of rewards delivered were analyzed separately. (supp Fig. 1a)”

line 6 page 9:

“The difference scores were not different from zero when an unexpected reward was delivered (Fig. 3e left) or when an expected reward was omitted (Fig. 3e right), regardless of whether the prediction error was induced by changes in timing or the number of rewards (supp Fig. 1b), reflecting the relatively flat firing in early and late trials of each type (Fig. 3f).”

2) Presence in well as hidden state?

If I understand correctly, the state-space task model assumes that the animal's presence in the left or right well is a hidden state (cf. Fig 5b - right). In other words, if not for their working memory, an animal would not know in which well their snout is located (left vs right). However, I assume that some visual and/or proprioceptive cues might make this state not completely hidden. Perhaps the authors could clarify and discuss this point (is

presence in port indeed considered a hidden state, and if so what are the limitations of this assumption).

Yes, our model assumes that the animal's presence in the left or right well is a hidden state. Moreover, every state is hidden in our model, and the model estimates the (hidden) states based on the observations, the previously estimated state, and other information. Most of the states in the flat model are signaled with a unique observation. Thus, these states are largely 'observable', not 'hidden'. However, the left and right well states share the same observations. At least the same external cues were shown in two states, making rats more challenging to differentiate. For the hierarchical model, four different states describe rats' presence in each well in four blocks, in which states are considered hidden for sure.

In addition, as we mentioned in our response to reviewer 1, we would like to clarify that we do not mean to suggest that the rats have no idea which well/block they are in. Instead, we suggest that the brain regions responsible for state estimation, the OFC and HC, do not function properly when lesioned and cannot utilize relevant information to estimate hidden states, resulting in dopamine calculating prediction errors inaccurately. Consequently, the model (or the related brain regions) was uncertain about the current well/block.

3) Time of hippocampus involvement in PEs computations

This study shows what type of hippocampus-dependent information is accessible to dopamine neurons for PE encoding. I'm curious about the timeline of this. Do the authors envision this hippocampal contribution happening on the fly, as PEs are computed (if so via which pathway)? Or is it the case that the hippocampus allows for a type of learning (possibly consolidated in the neocortex) that can then be called upon to compute PEs?

I don't necessarily expect the authors to have an answer, but it's something that the authors might want to discuss briefly.

As the reviewer may anticipate, we do not have a clear answer regarding when HC contributes to the learning process. Since the HC lesion was performed after rats had learned the task, it is less likely that the HC lesion disrupted the memory consolidation process, which typically happens early on. Thus, HC might play a more important role in online updating, which is necessary for each session. One possible mechanism, as we suggested in the manuscript, is that rats update the transition probability during the intertrial interval. This hypothesis is consistent with our knowledge that the replay/reactivation in the hippocampus, which is important for integrating information to understand the current context, occurs mostly when rats rest or at the event boundaries. With intact HC, the events that happened in the previous trial were replayed, and rats

estimated the current block with low uncertainty through replay/reactivation, while after the lesion, the estimations became more uncertain.

REVIEWERS' COMMENTS

Reviewer #1 (Remarks to the Author):

Thank you for the revisions. The model and experimental procedures are much clearer now. The inclusion of behavioral predictions from the hierarchical model is particularly useful.

Behavioral Predictions:

Please add to the discussion section the predictions the model would make for bilateral HCx lesions, as this could be valuable for guiding future research.

Reward Responsive Neuron Selection:

I am still struggling to convince myself that the selection of reward-responsive neurons and the findings are independent. Could the authors address this by:

- (a) Cross-validating neuron selection by performing the selection and the analyses shown in Fig 3 on different trials, not just time windows (if this hasn't been done already).
- (b) Showing what happens if neurons are defined based on a dip in response when comparing the 400 ms baseline from the inter-trial with reward omission trials.

Discussion of HPC and OFC Interaction:

The discussion of the interaction between HPC and OFC on page 21 is useful. It would be beneficial to include some previous relevant research on replay, such as Rusu et al. (2024), Kaefer et al. (2020), and Shuck & Niv (2019). Additionally, incorporating some of your own work on the interaction more generally, such as Wikenheiser et al. (2017) and Wang et al. (2020), would strengthen this section. Consider also referencing Mizrak et al. (2021) and Muhle-Karbe et al. (2023).

Overall, the revisions have improved the manuscript, and addressing these points would further enhance the robustness and comprehensiveness of the study. Thank you for your hard work.

Reviewer #2 (Remarks to the Author):

In a number of places the authors have been responsive to my comments. In others less so, but I'm not going to stand in the way of what is an interesting set of data.

Reviewer #3 (Remarks to the Author):

The authors did an excellent job of addressing my previous comments.

I have a question about the behavioral prediction of the hierarchical model (introduced in this revised manuscript). The authors show that rats are more likely to choose the high-valued well after receiving a big reward vs. after receiving a small reward. How is this prediction unique to the hierarchical model? Is there any indication that this behavioral pattern emerges only after extended training (after rats had a chance to learn the block-structure)?

Minor comments:

P12 L29: "The state space consisted of seven states: trial start, left/right cue, left/right well, left/right rewards (1st and 2nd drop), and inter-trial interval, as illustrated in Fig. 5a." Based on this description -and Fig 5b- I count ten states. Please clarify how many distinct states are included in the model.

Suppl. Fig.5: axis title is incorrect.

Please specify how many training sessions rats received before recording sessions. This is important since the assumption is that rats understood the block structure of the task.

We would like to express our gratitude to the reviewers for their support and constructive feedback. We have considered all reviewer's comments and made the necessary revisions. As a result, we believe that these revisions have significantly enhanced the quality of our manuscript. We have provided a detailed point-to-point response to the reviewers' comments, with their comments in black and our responses in blue. The major changes in the main manuscript are highlighted in yellow.

REVIEWERS' COMMENTS

Reviewer #1 (Remarks to the Author):

Thank you for the revisions. The model and experimental procedures are much clearer now. The inclusion of behavioral predictions from the hierarchical model is particularly useful.

Behavioral Predictions:

Please add to the discussion section the predictions the model would make for bilateral HCx lesions, as this could be valuable for guiding future research.

We added a paragraph discussing the bilateral HCx lesion.

Page 15, Line 20.

“Unilateral HC lesions did not result in significant behavioral changes, likely due to overtraining and the preservation of one intact hemisphere capable of maintaining normal behavior. This approach allowed us to investigate changes in DA firing resulting from HC output removal without behavioral confounds. Although we did not directly test the effects of bilateral hippocampal lesions on animal behavior in this study, previous research has also demonstrated that the hippocampus is crucial for context learning. Without it, animals can still learn but less efficiently. Consistent with these findings, our previous study, which manipulated reward size and type, found that rats with bilateral lesions of the hippocampal output area, the ventral subiculum, showed a reduced rate of choosing the higher-value reward in free-choice trials following block switches. without affecting choices in forced-choice trials.”

Reward Responsive Neuron Selection:

I am still struggling to convince myself that the selection of reward-responsive neurons and the findings are independent. Could the authors address this by:

(a) Cross-validating neuron selection by performing the selection and the analyses shown in Fig 3 on different trials, not just time windows (if this hasn't been done already).

We performed the suggested analysis to address potential selection bias. Reward-responsive dopamine neurons were selected by comparing their firing rates during baseline and reward epochs, excluding the first and last five trials ($n=37$ for control, 42 for HCx). We then examined the firing rates of these neurons specifically during those excluded trials.

As illustrated below, the results essentially replicate the findings in Figure 3 in the main text. In control animals, the difference scores were different from zero when an unexpected reward was delivered ($p=0.008$) and also when an expected reward was omitted ($p=3e-4$), reflecting the learning process. In HCx animals, as expected, neither difference score is significantly different from zero ($p=0.14$ for the reward delivery; $p=0.48$ for the reward omission). These results were consistent with our initial findings, indicating that our neuron selection method did not bias the results.

Fig1. Reward-evoked changes in activity of cross-validated reward-responsive dopamine neurons. (a and b) Population responses of reward-responsive dopamine neurons in Control (a) and HCx (b) groups. (c and e) Distributions of difference scores comparing firing to unexpected reward delivery (left) and omission (right) in the early and late trials in control (c) and HCx (e) groups. The numbers in each panel indicate results of two-sided Wilcoxon signed-rank test (p) and

the average difference score (u). **(d and f)** Average firing in response to reward delivery (black) and omission (gray) in the first 5 and last 5 trials in control (d) and HCx (f) groups.

(b) Showing what happens if neurons are defined based on a dip in response when comparing the 400 ms baseline from the inter-trial with reward omission trials.

As requested above, we used the neuronal response to reward omission as a criterion for identifying dopamine neurons for the analysis presented in Figure 3. To be clear, we do not know the purpose of this selection criterion, but as it was requested, we have done it. As expected, this approach yielded a very small sample of neurons ($n=15$ for control, 9 for HCx). Both because of the small number of neurons and also because the analysis is cherry picking neurons that show reduced firing on reward omission, both controls and HCx groups looked different from what was in the text. The results lack statistical power, thus we show them below but have not included them in the main text.

Fig2. Reward-evoked changes in activity of dopamine neurons responsive to reward omission. **(a and b)** Population responses of reward-responsive dopamine neurons in Control **(a)** and HCx **(b)** groups. **(c and e)** Distributions of difference scores comparing firing to unexpected reward delivery (left) and omission (right) in the early and late trials in control **(c)** and HCx **(e)** groups. The numbers in each panel indicate results of two-sided Wilcoxon signed-rank test (p) and the average difference score (u). **(d and f)** Average firing in response to reward delivery (black) and omission (gray) in the first 5 and last 5 trials in control **(d)** and HCx **(f)** groups

Discussion of HPC and OFC Interaction:

The discussion of the interaction between HPC and OFC on page 21 is useful. It would

be beneficial to include some previous relevant research on replay, such as Rusu et al. (2024), Kaefer et al. (2020), and Shuck & Niv (2019). Additionally, incorporating some of your own work on the interaction more generally, such as Wikenheiser et al. (2017) and Wang et al. (2020), would strengthen this section. Consider also referencing Mizrak et al. (2021) and Muhle-Karbe et al. (2023).

We have included these relevant studies in our discussion. The changes are listed below.

Page 14, Line 9. “While prior studies (Mizrak, et al. 2021; Muhle-Karbe, et al. 2023) have emphasized similarities in task space encoding between the HC and OFC, the current findings reveal important distinctions.”

Page 15, Line 4. “Without an intact HC and the reactivation of HC neurons, the estimation of the block may become more uncertain. It is worth noting that while similar reactivation patterns have been observed in other brain regions, including the medial prefrontal cortex (Kaefer, et al. 2020) and OFC (Rusu, et al. 2024), they seems to be insufficient to fully compensate for HC function.

Page 15, Line 20. “Unilateral HC lesions did not result in significant behavioral changes, likely due to overtraining and the preservation of one intact hemisphere capable of maintaining normal behavior. This approach allowed us to investigate changes in DA firing resulting from HC output removal without behavioral confounds. Although we did not directly test the effects of bilateral hippocampal lesions on behavior in this study, previous research (Langston and Wood 2010; Wiltgen, et al. 2006) has demonstrated that the hippocampus is crucial for context learning. Without it, animals can still learn but less efficiently. Consistent with these findings, our previous study (Wikenheiser, et al. 2017), which manipulated reward size and type, found that rats with bilateral lesions of the hippocampal output area, the ventral subiculum, showed a reduced rate of choosing the higher-value reward in free-choice trials following block switches. without affecting choices in forced-choice trials.”

Page 16, Line 1. “Although we have shown the distinct contributions of the OFC and HC to the cognitive map, how they interact and contribute collectively remains a mystery. Our previous study (Wikenheiser, et al. 2017) suggested that the inactivation of HC impaired the representation of task space in OFC. The current findings shed further light on this complex interaction.”

Overall, the revisions have improved the manuscript, and addressing these points would further enhance the robustness and comprehensiveness of the study. Thank you for your hard work.

We are glad to hear that the revisions have improved the manuscript, and thank you for acknowledging our work. We believe we have addressed the remaining points and further enhanced our study.

Reviewer #2 (Remarks to the Author):

In a number of places the authors have been responsive to my comments. In others less so, but I'm not going to stand in the way of what is an interesting set of data.

We appreciate your compliment on our work and are glad we could address many of your comments.

Reviewer #3 (Remarks to the Author):

The authors did an excellent job of addressing my previous comments.

I have a question about the behavioral prediction of the hierarchical model (introduced in this revised manuscript). The authors show that rats are more likely to choose the high-valued well after receiving a big reward vs. after receiving a small reward. How is this prediction unique to the hierarchical model? Is there any indication that this behavioral pattern emerges only after extended training (after rats had a chance to learn the block-structure)?

We apologize that this was confusing. This prediction is unique to the hierarchical model because it can use the overall task structure to make predictions across wells. In the hierarchical model, receiving a big or small reward provides different amounts of information about the reward outcome in the other well. When an immediate/small reward is received from one well, the model predicts the other well contains either a big or delayed reward, which could be either more or less advantageous. However, when a big or delayed reward is received from one well, the model confidently predicts an immediate/small reward in the other well, eliminating uncertainty about the reward outcome associated with two wells. Therefore, the hierarchical model is able to make better judgments about which well is associated with a better reward after receiving a big or delayed reward, than receiving an immediate/small reward.

The flat model, in contrast, lacks this capability. It cannot make cross-well predictions based on the overall task structure, instead treating each well independently without inferring relationships between them. Consequently, the flat model would not be

expected to show any difference in predictions after receiving a big reward versus after receiving a small reward.

The point about testing rat behavior at the start of training is insightful, but unfortunately, our training procedure does not support such analysis. Rat training proceeded in three phases: First, rats had five sessions of free-choice trials with equal, constant rewards in both wells. In this phase, if a rat showed a strong side bias, one side was physically and temporarily blocked to encourage balanced exploration. Then, they underwent at least ten sessions of forced-choice trials, using the recording session reward configuration, until reaching an 80% correct rate. Finally, we combined free-choice and forced-choice trials. Consequently, rats only encountered free-choice trials with unequal rewards after they had already learned the block structure and achieved high performance. This approach prevented us from analyzing rat behavior in free-choice trials when they were still unfamiliar with the task structure.

The changes were made as follows

Supp. page 7, line 2:

“After switching to the 2nd and 4th blocks, rats received unexpected rewards at both wells. In the flat model, these unexpected rewards only modify the value of entering that well, while in the hierarchical model, an unexpected reward in one well can modify the value of reward in the other well, using knowledge of the task structure. However, this knowledge is not unambiguous. Due to quirks in the block design, when an immediate or small reward is received, the model predicts that the other well contains either a big or a delayed reward, which leaves some ambiguity since these two rewards can be either more or less advantageous. By contrast, when a big or delayed reward is received, the model confidently predicts an immediate/small reward in the other well, eliminating uncertainty about the reward outcomes associated with both wells. Therefore, the hierarchical model is able to make better judgments about which well is associated with a better reward after receiving a big or delayed reward, than after receiving an immediate/small reward. In contrast, the flat model cannot make cross-well predictions based on the task structure, instead treating each well independently without. Consequently, the flat model would not be expected to show any difference in predictions after receiving a big reward versus after receiving a small reward.”

Minor comments:

P12 L29: "The state space consisted of seven states: trial start, left/right cue, left/right well, left/right rewards (1st and 2nd drop), and inter-trial interval, as illustrated in Fig.

5a." Based on this description -and Fig 5b- I count ten states. Please clarify how many distinct states are included in the model.

There are ten states, sorry for the mistake. We have updated it.

Suppl. Fig.5: axis title is incorrect.

Fixed

Please specify how many training sessions rats received before recording sessions. This is important since the assumption is that rats understood the block structure of the task.

Thanks for the suggestion, and we have added this in the method section.

page 19, line 11:

“Rat training proceeded in three phases. Initially, rats underwent five sessions with only free-choice trials, where rewards in both wells were equal and constant. If a rat showed a strong side bias, one side was temporarily blocked to encourage balanced exploration. Next, rats experienced at least ten sessions of forced-choice training, using the same reward configuration as in recording sessions. This phase continued until rats achieved an 80% correct rate. Finally, free-choice and forced-choice trials were combined. The entire training lasted approximately 25 sessions and all rats could reliably complete all four trial blocks in a single session before the recording sessions began.”